# Can You Make Better Decisions If You Are Bilingual?

**Alena Kirova [1] and Jose Camacho [2],***

1. Department of English and World Languages, Youngstown State University, Youngstown, OH 44555, USA; akirova@ysu.edu
2. Department of Hispanic and Italian Studies, University of Illinois Chicago, Chicago, IL 60607, USA
* Correspondence: jcamach@uic.edu

**Abstract:** Studies have shown that "framing bias," a phenomenon in which two different presentations of the same decision-making problem provoke different answers, is reduced in a foreign language (the Foreign Language effect, FLe). Three explanations have emerged to account for the difference. First, the *cognitive enhancement hypothesis* states that lower proficiency in the FL leads to more deliberate processing, reducing the framing bias. Second, contradicting the previous, the *cognitive overload hypothesis* states that the cognitive load actually induces speakers to make less rational decisions in the FL. Finally, the *reduced emotionality hypothesis* suggests that speakers have less of an emotional connection to a foreign language (FL), causing an increase in rational language processing. Previous FLe research has involved both FL and non-FL speakers such as highly proficient acculturated bilinguals. Our study extends this research program to a population of heritage speakers of Spanish (HS speakers), whose second language (English) is dominant and who have comparable emotional resonances in both of their languages. We compare emotion-neutral and emotion-laden tasks: if reduced emotionality causes the FLe, it should only be present in emotion-laden tasks, but if it is caused by cognitive load, it should be present across tasks. Ninety-eight HS speakers, with varying degrees of proficiency in Spanish, exhibited cognitive biases across a battery of tasks: framing bias appeared in both cognitive-emotional and purely cognitive tasks, consistent with previous studies. Language of presentation (and proficiency) did not have a significant effect on responses in cognitive-emotional tasks, but did have an effect on the purely-cognitive Disjunction fallacy task: HS speakers did better in their second, *more* proficient language, a result consistent with neither the reduced emotionality hypothesis nor the cognitive enhancement hypothesis. Moreover, *higher* proficiency in Spanish significantly improved the rate of *correct* responses, indicating that our results are more consistent with the cognitive overload hypothesis.

**Keywords:** foreign language effect; framing bias; decision-making; heritage language

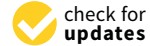



## 1. Introduction

When people make certain kinds of decisions, they are systematically sensitive to the way in which information is presented (Tversky and Kahneman 1981). For example, when a task presents two alternatives to choose from in which some kind of gain is involved, the majority of participants tend to be conservative and select the choice that involves less potential risk. However, when the same task is presented as involving a loss, participants tend to select the riskier choice. This tendency is irrational, since the way a task is presented/framed should not have an effect on the decision. This difference is known as a framing bias or framing effect (Tversky and Kahneman 1986, 1991; Kahneman 2003, 2011; Kahneman and Frederick 2007).

Subsequent research indicates that when a similar decision task is presented in a foreign language, this bias is reduced or eliminated (Keysar et al. 2012; Costa et al. 2014a; Costa et al. 2017; Ivaz et al. 2019; see Polonioli 2018 for critical assessment)—a finding that has been referred to as Foreign Language Effect (FLe). Keysar et al. (2012) tested three separate populations of L2 speakers—English-L1/Japanese-L2, Korean L1/English-L2 and

English-L1/French-L2—and found a reduced bias when the decision-making tasks were presented in the L2. In a follow-up study, Costa et al. (2014a) replicated the finding and extended it to a number of different decision-making tasks in speakers of Spanish-L1/English-L2, Arabic-L1/Hebrew-L2, and English-L1/Spanish-L2. In addition to these two studies on framing effect, a FLe has been found in a number of studies on moral decision-making: people scored higher on their tendency to maximize benefit when presented with dilemmas in their FL than in their L1 (Costa et al. 2014b; Geipel et al. 2015), a finding referred to as a moral Foreign Language effect (MFLe).

Three hypotheses have been put forward to account for the FLe both on cognitive biases and moral decision-making. The three hypotheses can be traced back to theories on the perception of risk and benefit, according to which judgment and decision-making happen via two routes or systems (e.g., Loewenstein et al. 2001; Slovic et al. 2004): an intuitive, quick, and automatic route called System 1 and an analytical, slow, and cognitively effortful route called System 2. A decision made via the first route is considered to be based on *affect heuristic*, where the positive or negative associations activated by the description of the task predetermine the decision (Damasio 1994; Slovic et al. 2002). By contrast, a decision made via the second route is considered to be based on normative principles (e.g., expected utility theory) and on analysis and prediction of potential outcomes.

Specifically, the first hypothesis, the reduced emotionality hypothesis (REH), rests on the idea that L2 words have a weaker and less automatic emotional effect on people, an idea supported by a number of clinical, cognitive, psychophysiological, and neuroimaging studies (for a comprehensive review, see Pavlenko 2012; Caldwell-Harris 2014). According to this hypothesis, the weakened emotional effect of the L2 words inhibits the quick affect-based response generated by System 1, leading to less emotional decisions. For example, people's negative reactions to a harm-causing action (e.g., killing a person to save five) are "blunted" when presented as a moral dilemma in a FL because they are less affected by the emotional component of the dilemma, thereby allowing them to focus on the benefit-maximizing component of the action rather than on causing harm. Similarly, the linguistic description of distressing life-death scenarios in a framing bias scenario such as the Asian disease task does not trigger negative affect to the same extent when presented in the FL, thereby decreasing framing bias. In other words, thinking about saving lives or causing deaths in one's FL is not as emotional as it is in the L1, hence is less prone to affect-based decision-making (Keysar et al. 2012; Costa et al. 2014a on cognitive biases; and Cipolletti et al. 2016; Geipel et al. 2015; Costa et al. 2014b on moral dilemmas).

The *cognitive enhancement hypothesis* (CEH) is an alternative to the reduced emotionality hypothesis. This hypothesis explains the FLe by positing that the greater cognitive load involved in FL processing forces one to slow down, thus inhibiting the quick spontaneous response generated by System 1 (Hayakawa et al. 2017). As a result, processing is "routed" to System 2, which promotes a more analytical approach to decision-making (Keysar et al. 2012), thereby *enhancing* responses to decision-making problems, both for moral decision-making (Hayakawa et al. 2017) and framing biases (Costa et al. 2014b).

The third hypothesis, which in this study we will refer to as the *cognitive overload hypothesis* (COH), is the exact opposite of the cognitive enhancement hypothesis in that it states that decision-making in one's FL actually exacerbates biases by overloading the processor. According to Costa et al. (2014a, pp. 238–39), "Under conditions of high cognitive load participants' decisions tend to be more affected by heuristic biases (Benjamin et al. 2006; Whitney et al. 2008; Forgas et al. 2009). That is, when cognitive load taxes System 2, the rational processor cannot check or control the intuitive answers given by System 1. Hence, to the extent that reading in a FL increases cognitive load, one might expect heuristic biases to affect participants' responses to a larger extent when the problem is set in a FL." Hayakawa et al. (2017) make a similar prediction about FLe on moral decision-making.

In previous studies, the FLe has been mostly tested using FL learners (Keysar et al. 2012; Costa et al. 2014a on framing biases; Costa et al. 2014b; Geipel et al. 2015; Hayakawa et al.

2017; Mills and Nicoladis 2020; Białek et al. 2019 on moral decision-making). Following Gass et al. (2013), foreign language learning is defined as the study of a second language in a formal classroom situation that takes place in a country where the native language is dominant. Crucially, for such FL learners the FL is both the less emotional *and* the less proficient language, making it difficult to tease apart the effects of reduced emotionality and cognitive enhancement/overload. Furthermore, the limited populations tested in such studies may occlude the possibility that the FLe is a characteristic unique to FL learners and may be absent in other language populations. Thus, one of the purposes of this article—along with others that test the FLe in, for example, highly proficient acculturated bilinguals, who are not foreign language learners in the traditional sense of the term—is to test the limits of the FLe. This is an important point, because FL speakers represent just a fraction of the world's bilingual population. Bilingualism encompasses a large and diverse set of speakers, including balanced bilinguals, unbalanced bilinguals who have learned their L2 in a context other than the classroom, and heritage speakers, and conventional studies on FLe cannot clarify what (if any) kind of language effect is present in those bilingual populations.

More recently, a number of studies have used highly proficient acculturated bilinguals instead of FL learners to test the FLe (Čavar and Tytus 2018; Brouwer 2019, 2020; Dylman and Champoux-Larsson 2020; Miozzo et al. 2020). The results are illuminating, but inconclusive. For example, Brouwer (2019) found no language effect on moral decision-making in highly proficient Dutch-English bilinguals, but Brouwer (2020) did. Čavar and Tytus (2018) also failed to find a language effect on moral decision-making in highly-acculturated Croatian-German bilinguals (but see Białek and Fugelsang 2019 for critique of their conclusions and Krautz and Čavar 2019 for their responses). While Dylman and Champoux-Larsson (2020) found language effect on neither moral decision-making nor framing in highly acculturated Swedish-English bilinguals, Miozzo et al. (2020) did find a language effect on moral dilemmas and framing bias in native Italian-Venetian and Italian-Bergamasque bilinguals.

These studies shed more light on the issue of FLe origin and scope, but while highly proficient acculturated bilinguals in those studies are not FL learners, their first language is still always the dominant one, and thus both more proficient as well as more emotional. Therefore, we complement data from the studies on FL learners and highly acculturated bilinguals with data from a novel and principally different population: heritage speakers. Heritage speakers represent a unique language population that, to the best of our knowledge, has not been tested in the studies investigating FLe on judgment and decision-making. For the purposes of this article, we adopt the definition of heritage speakers as "individuals raised in homes where a language other than English is spoken and who are to some degree bilingual in English and the heritage language" (Valdés 2000). While heritage speakers may come from different countries, we only recruited heritage speakers of Spanish (HS speakers) in our study, because we did not want to introduce additional variables (e.g., Spanish-speaking country of origin vs. Russian-speaking country of origin, etc.) in the design.

There are two principal differences between FL learners/highly proficient bilinguals in the previous studies and the HS speakers in our study. First, *proficiency* in heritage speakers' L1 vs. L2 language is reversed compared to typical bilinguals —HS speakers are typically *less proficient* in their L1 than in their L2. This pattern is caused by their language acquisition history: they are born into Spanish speaking families and learn Spanish as their L1, but are then exposed to and learn English (L2) because of societal and schooling needs, and receive formal education in their L2 but generally not in their L1. As a result, English becomes their dominant and more proficient language.

Second, *emotional resonance*, which "refers to the emotionality elicited by a given problem" (Costa et al. 2014a, p. 237), should be comparable in HS speakers' two languages, because both languages are learned in an immersion context—the L1 with family, and the L2 with friends, schoolmates, etc. By contrast, FL learners study their FL in a classroom

setting and thus should have a weaker emotional connection to it. In this study, we measure the perceived "emotionality" of Spanish and English in HS speakers using an adapted version of the Emotional Phrases Task from Caldwell-Harris and Ayçiçegi-Dinn (2009). We expect to find no differences between Spanish and English, because a number of studies have compared early and late bilinguals and found that the reduced emotionality effect is present in late bilinguals, but it diminishes or disappears in early bilinguals (Anooshian and Hertel 1994; Harris 2004; Harris et al. 2006; Sutton et al. 2007; Eilola and Havelka 2011; Ferré Pilar et al. 2010; Caldwell-Harris et al. 2011; Ferré et al. 2018; Ivaz et al. 2019; Miozzo et al. 2020). That is, if the L2 is acquired earlier in life and in a naturalistic setting, it evokes emotional resonances similar to the L1. For example, Spanish–English early bilinguals who had learned their L2 (English) before puberty and often in a naturalistic environment (studied in the USA and used English regularly) in Ferré Pilar et al. (2010) recalled emotional words at the same rate in L1 and L2. Harris (2004) compared 31 early Spanish-English bilinguals who were born in the US or immigrated to the US before the age of 7 and 21 late bilinguals who arrived in the US at or after the age of 12. Early bilinguals rated themselves as either balanced bilinguals or dominant in English, while late bilinguals' most proficient and likely most dominant language was Spanish. Results showed that L1 and L2 had similar emotional strength for early bilinguals, and the author concluded that the L1 is only perceived to be more emotional if it is the more proficient language.

Similarly, in Caldwell-Harris et al. (2011), late Mandarin–English bilinguals rated the L1 Mandarin to be more emotional, but this language effect disappeared in early bilinguals, who rated the two languages as equally emotional. Moreover, Caldwell-Harris et al. (2012) explored perceived language emotionality of L1 Russian L2 English speakers who arrived in the USA at different ages or learned Russian as their second language. A comparison of three groups—Russian native speakers who arrived to the USA before the age of 10 (early arrivals), Russian native speakers who arrived to the USA after the age of 10 (late arrivals), and L1 English speakers who learned Russian as a foreign or second language—showed that perceived emotionality of Russian was the highest for late arrivals, followed by early arrivals, and was the lowest in the L2 group.

In a recent behavioral study, Ferré et al. (2018) administered a lexical decision task (LDT) and an affective decision task (ADT) to highly proficient balanced Catalan-Spanish bilinguals in both of their native languages, and to a group of Catalan-Spanish bilinguals in their FL English. Language effect was found only in the FL group (when the tasks were performed in English), but not in the Catalan-Spanish groups. Since Catalan and Spanish were acquired in early childhood in a naturalistic environment, while English was studied as a foreign language, these results support the idea that languages acquired early in life have comparable emotional resonances. The authors also suggest that FL words are not as grounded in sensorimotor experiences as L1 words.

Also relevant is Ivaz et al. (2019) on self-bias and non-nativeness vs. foreignness. The authors employed self-bias paradigm to establish whether the foreign language effect came from non-nativeness or foreignness of a language. They tested Spanish native speakers who were born and raised in the Basque Country and who spoke Basque and English with relatively high and approximately equal proficiency. Crucially, while both Basque and English are not their native languages, the participants had learned the former in an immersion context by virtue of living and working among people who speak it as a native tongue, while they learned English in a largely impersonal and unemotional formal classroom context. The results showed a reduction of self-bias in the non-native foreign (English) language, but not in the non-native local language (Basque), which indicates that the FLe is caused by foreignness, not non-nativeness of a language.

In fact, recent discoveries in neuroscience research indicate that, provided the right context, one may be able to continue building emotional resonances in the L2 well into adulthood. Sorrells et al. (2019) found a group of neurons in the paralaminar nuclei (PL) of the human amygdala—the center of emotional processing in the brain—that remain

immature late into adulthood. Most of these neurons rapidly mature in adolescence, which can account for the tumultuous development of emotional intelligence in teenagers, but some of them remain immature throughout life, thus possibly allowing the brain to remain flexible as far as emotional processing is concerned (Sorrells et al. 2019). This suggests that the reason the L2 is typically less emotional does not have to do with some kind of neurological maturational constraint but rather with the absence of emotional stimulation. Most importantly for the purposes of this study, it suggests that our HS speakers, who were exposed to their L2 at an average age of 4.9 and learned it in a naturalistic environment, should have been able to build similar emotional resonances in their L1 and L2.

Thus, since HS speakers' second language is almost always the more proficient one, and, presuming that they have comparable emotional resonances in both of their languages, the population should be distinct from the FL learners and highly proficient acculturated bilinguals in the recent studies on Moral FLe (see Table 1). Studying this unique population allows us to tease apart the effects of emotionality and cognitive load.

**Table 1.** Differences between bilingual populations.

| Population | More Emotional in | More Proficient in |
|---|---|---|
| FL learners | L1 | L1 |
| Highly proficient acculturated bilinguals | typically L1 | typically L1 |
| Heritage speakers | neither | typically L2 |

If the FLe is caused by reduced emotionality in the FL (the reduced emotionality hypothesis), we would not expect to observe a language-associated reduction of decision-making biases in heritage speakers, since they have similar emotional resonances in both of their languages. Alternatively, if the FLe is caused by cognitive enhancement (the cognitive enhancement hypothesis) due to a more deliberate processing caused by the lower proficiency in the L1 (Spanish), the FLe should be present in HS speakers' less proficient L1, in contrast with the typical L2 populations where it is present in the less proficient L2. Finally, if the cognitive overload hypothesis is right and lower proficiency in a language actually exacerbates rather than reduces decision-making biases, the bias reduction should be present in the HS speakers' more proficient L2, since higher proficiency in the L2 (English) should lead to more rational results.

In addition to complementing the existing data on the FLe with data from a novel population of HS speakers, we also complement it with data from emotion-laden vs. emotion-neutral tasks. Task type is an important variable when considering the FLe for the following reasons: if the FLe is caused by reduced emotionality in the foreign language, it should only be present in tasks that involve emotionality, but if it is caused by cognitive load, it should be present in any task that involves cognitive processing. Therefore, in order to establish whether the FLe stems from reduced emotionality or cognitive enhancement/cognitive overload it is critical to employ tasks that involve both emotional and emotionally-neutral problems. Costa et al. (2014a) and Vives et al. (2018) included such problems in their studies of FL learners, but the recent studies that specifically looked at the FLe in highly proficient acculturated bilinguals employed only emotion-laden tasks. For example, Miozzo et al. (2020) and Dylman and Champoux-Larsson (2020) used the Asian Disease Problem and the Footbridge Dilemma, Brouwer (2020) used personal and impersonal dilemmas, and Čavar and Tytus (2018) used a number of moral dilemmas from Bartels (2008). In other words, as of yet early bilinguals have not been tested on unemotional tasks.

For that reason, in this study we tested our HS speakers on both emotion-laden and emotion-neutral tasks. The former may trigger negative affect (unpleasant emotions such as anxiety, fear, shame, guilt, irritability, etc.) (Watson et al. 1988), while the latter typically do not produce such an effect on people. Examples of emotion-laden tasks include those involving risk aversion and loss aversion, moral dilemmas, etc., while examples

of emotion-neutral tasks include those testing the outcome bias, the conjunction fallacy, disjunction fallacy, base-rate neglect fallacy, the cognitive reflection test, etc. We refer to the former as "cognitive-emotional" problems and to the latter as "purely-cognitive" problems. Recruiting the population of HS speakers and employing these two types of tasks allows us to test the three hypotheses—reduced emotionality, cognitive enhancement, and cognitive overload—and make the following predictions. First, if the reduced emotionality hypothesis is correct, and given the above-mentioned findings showing comparable emotionality in the two languages of early bilinguals, there should neither be language-associated bias reduction in cognitive-emotional nor in purely-cognitive tasks in our HS speakers. Second, if the cognitive enhancement hypothesis is correct, HS speakers should display language-associated bias reduction in their less proficient L1 (Spanish), where their lack of proficiency would make them slow down and give a more deliberate response across the tasks (because all tasks involve some kind of cognitive effort). This is the opposite of the typical FL populations where the bias reduction is present in the FL, because the FL is always less proficient in such populations. Third, if the cognitive overload hypothesis is correct, bias reduction should be found in the more proficient L2 across the tasks, unlike in the typical FL populations where it should be present in the more proficient L1. Table 2 summarizes these hypotheses, and the present study will allow us to test them with respect to the HS speakers (rightmost column of Table 2).

**Table 2.** Hypotheses regarding the Foreign Language Effect (FLE).

| Hypothesis | FL Learners (Previous Studies) | Heritage Speakers (Current Study) |
| --- | --- | --- |
| Reduced emotionality | Bias-reduction in less emotional L2 | Since both equally emotional, bias-reduction in neither language in any of the tasks |
| Cognitive enhancement | Bias-reduction in less proficient L2 | Bias-reduction in less proficient L1 (Spanish) across all tasks |
| Cognitive overload | Bias-reduction in more proficient L1 | Bias-reduction in more proficient L2 (English) across all tasks |

Complementing existing research on FL learners and highly-proficient bilinguals, our study of HS speakers will provide additional insights into the origin and scope of the FLe, explore language effects on a language population other than FL learners, and will make it possible to tease apart the effects of higher emotionality and higher proficiency. Therefore, in this paper, we contribute to the discussion of the FLe by extending the decision-making experiments to bilingual heritage speakers of Spanish (HS) who are dominant in English, as well as employing decision-making tasks that both involve an emotional component and those that do not. In addition, ours is the first study after Costa et al. (2014a) to explore the FLe on several cognitive biases rather than moral dilemmas.

## 2. Materials and Methods

### 2.1. Participants

Ninety-eight heritage speakers of Spanish were recruited among Rutgers University students for course credit. Four were removed from the final statistics because they listed Spanish as their dominant language. The remaining 94 ranged in age between 18–49 (mean age = 21; *SD* = 3.6; 64 females). All participants gave a written informed consent to participate in the study, and the study was approved by the Rutgers University Institutional Review Board (IRB).

Participants' average age of onset of exposure to English was 4.9 years old (*SD* = 3.53). They filled out a linguistic history questionnaire (see Appendix A), summarized in Table 3. All of the participants were exposed to Spanish from 0–5 years of age, and most of them were also exposed to English. According to their self-report, during the ages 0–5 they were exposed to Spanish 60% of the time vs. English 40%. Their schooling was done mostly in English through secondary school and beyond (73%), their time spent in English vs.

Spanish-speaking countries was 78% vs. 22% respectively, and their own preference in addressing someone who speaks both English and Spanish was 72% vs. 28% respectively, all suggesting that their dominant social language is English.

**Table 3.** Average responses to background questionnaire.

| Percentages | Spanish | English |
|---|---|---|
| Years spent in country speaking | 22% | 78% |
| Years of schooling in | 27% | 73% |
| Language exposure (0–5 yrs) | 60% | 40% |
| Language preference with someone who speaks both languages | 28% | 72% |

The suggestion that English is their dominant language is confirmed by self-rating of their ability to write, read, listen and understand English and Spanish on a scale of 0 to 5. Ratings for all four categories were averaged per participant, and then averaged for English (4.8, SD = 0.42), and for Spanish (4.15, SD = 0.72). The difference was statistically significant (paired *t*-test: $t(93) = 7.66$, $p < 0.001$, $d = 0.8$), suggesting that English is self-perceived as the more proficient language. Based on these self-ratings, we eliminated four participants from the resulting statistics, as described earlier.

In addition to the self-reports, Spanish proficiency was measured through a standardized Spanish task, first used in Duffield and White (1999). This test incorporates the reading and vocabulary sections of the MLA Cooperative Foreign Language Test (Educational Testing Service, Princeton, NJ) and parts of the Diploma de Español como Lengua Extranjera (DELE). The test consists of 50 grammar and vocabulary questions in multiple-choice and fill-in-the-blank formats. Proficiency, which was used as a continuous variable, ranged between 13–96% (*M* = 76.7%, *SD* = 18.1). Despite a relatively high self-rating in Spanish reported above, participants' average result in the proficiency task was 76.7%, suggesting that Spanish proficiency is low on average.

*2.2. Sentence Ratings for Emotionality*

We adopted the Emotional Phrases Task from Caldwell-Harris and Ayçiçegi-Dinn (2009) to measure perceived "emotionality" of Spanish and English among heritage speakers of Spanish. The task consists of six endearments (e.g., "You are everything to me"), six insults (e.g., "You are so ugly,"), six reprimands (e.g., "Don't talk back!"), and ten emotionally neutral items. The English stimuli adopted from Caldwell-Harris and Ayçiçegi-Dinn (2009) were translated into Spanish by a native speaker of Spanish and the translations were checked by two Spanish-English bilinguals. To ensure that the thirty original English sentences and their Spanish translations were comparable in terms of emotionality, we ran a norming study with thirty native speakers of Spanish and thirty native speakers of English. These speakers rated emotionality of the English and Spanish sentences in their respective native language on a 5-point Likert scale where 0 meant not at all emotional and 4 meant very emotional. Eight sentences were rated significantly differently in the two languages and were thus discarded. As a result, our adapted version of the Emotional Phrases Task consisted of 22 phrases, half of which were "emotional" and half "unemotional" (see Appendix B).

Our heritage speaker participants were randomly assigned to rank emotionality of these phrases either in their L1 (Spanish) or L2 (English) using the same 5-point Likert scale used for the norming study. The stimuli were presented in written modality, since we wanted to avoid the possibility that minor differences in intonation and pitch inevitable in oral modality would affect the ratings. Figure 1 seems to indicate that less emotional phrases were rated higher in the L1 (Spanish) on average and more emotional ones had more mixed ratings. However, a 2 (Language: Spanish vs. English) × 2 (Emotionality: Emotional vs. non-emotional phrases) repeated measures ANOVA revealed a significant effect of Emotionality on ratings, $F(1,84) = 533.885$, $p < 0.05$, but no effect of Lan-

guage, $F(1,84) = 2.071$, $p = 0.154$, and no interaction between Language and Emotionality, $F(1,84) = 1.141$, $p = 0.289$. These results show that emotional ratings are not significantly different in both languages, and suggest that our participants treated both languages non-distinctly in terms of emotionality.

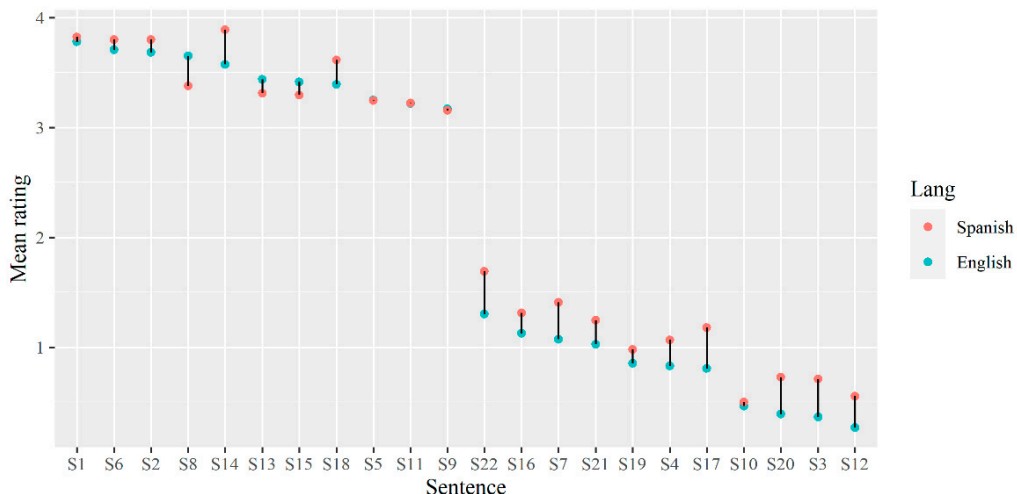

**Figure 1.** Average rating for phrases by language.

*2.3. Procedure*

Decision-making tasks in this study belong to two types: ones that require only cognitive processing and ones that require cognitive processing while at the same time potentially evoking an emotional reaction (Costa et al. 2014a; Vives et al. 2018). In this study we refer to the former as "purely-cognitive" problems (Cognitive Reflection Test and Disjunction fallacy problem) and to the latter as "cognitive-emotional" problems (risky-choice framing, decision-making under risk and uncertainty, and psychological accounting of economic outcomes).

*2.4. Tasks*

2.4.1. Cognitive-Emotional Decision-Making Tasks

These tasks included risky-choice framing tasks (Asian disease problem and Financial crisis problem), psychological accounting of economic outcomes (Money lost vs. Ticket lost problem), and decision-making under risk and uncertainty (Holt-Laury Lottery).

Risky-choice framing: We used the classic Asian disease problem and its less famous alternative called the Financial crisis problem to test the risky-choice framing effect in heritage speakers. Framing effect refers to people's tendency to select the same option more often when it is described using positive semantics than when it is described using negative semantics (Tversky and Kahneman 1981). For example, in the Asian disease problem participants are presented with the following description:

Recently, a dangerous new disease has been going around. Without medicine, 600,000 people will die from it. In order to save these people, two types of medicine are being made.

They need to choose one of the two potential solutions that are framed as either saving lives (gain frame) or losing lives (loss frame):

Gain frame presentation:

(A) If you choose Medicine A, 200,000 people will be saved.

(B) If you choose Medicine B, there is a 33.3% chance that 600,000 people will be saved and a 66.6% chance that no one will be saved.

Loss frame presentation:

(C) If you choose Medicine A, 400,000 people will die.

(D) If you choose Medicine B, there is a 33.3% chance that no one will die and a 66.6% chance that 600,000 people will die.

Options A and C are numerically identical, and so are options B and D; however, people choose the safe option A reliably more often in the gain frame, and choose the gamble option D in the loss frame, thereby exhibiting a framing effect (Tversky and Kahneman 1981).

The Financial crisis problem is equivalent to the Asian disease problem, but it involves people losing jobs rather than lives. This problem is arguably less emotional, but it still involves an emotional component and has been used in previous literature to test framing effects (Costa et al. 2014a; Liberman et al. 2004).

Psychological (or mental) accounting of economic outcomes: When people make decisions about how to spend or invest their money, they engage in mental accounting, which is defined as " . . . the set of cognitive operations used by individuals and households to organize, evaluate and keep track of financial activities" (Thaler 1999). Mental accounting is local instead of global and thus irrational, because people tend to mentally compartmentalize their overall budget into arbitrary subjective (local) categories such as "grocery money", "money for fun", "money for utilities", etc., and make decisions about spending the money based on the category. For example, they will be more likely to treat a tax refund as "found money" and thus spend it on a vacation or expensive gift instead of using it for more practical purposes. This violates the *fungibility* (substitutability) principle—the fact that all money within a budget is entirely substitutable (Thaler 1999).

We used a version of the Ticket lost/Money lost problem to test the effect (if any) of language of presentation on mental accounting. The task measures whether people make decisions globally or locally. Participants were presented with either the ticket lost option or the money lost option and had to indicate whether they would or would not buy the tickets.

Ticket lost option:

You have bought two tickets to go to the theatre. Each ticket costs $80. When you arrive at the theatre, you open your bag and discover that you have lost the tickets.

Would you buy the tickets to enter the theatre?

Money lost option:

You go to the theatre and want to buy two tickets that cost $80 each. You arrive at the theatre, open your bag, and discover that you have lost the $160 with which you were going to buy the tickets.

Would you buy the tickets to enter the theatre?

People are reliably less likely to buy the tickets in the ticket lost option, because they feel that they have depleted their "theatre" funds and don't have the right to draw from other funds to buy the ticket. In the money lost option, on the other hand, they are more willing to buy the tickets, because they don't consider loss of money part of the cost of tickets (Liberman et al. 2004; Costa et al. 2014a), thereby exhibiting irrational, local, mental accounting (Tversky and Kahneman 1981; Costa et al. 2014a).

While these mental accounting problems do not involve risk aversion or loss aversion in their classical sense, they may be considered emotional due to the necessity to make decisions about losing/spending the participant's money.

Decision-making under risk and uncertainty: the Holt-Laury lottery (Holt and Laury 2002) is a test designed to measure risk aversion under uncertainty and represents another decision-making task that involves emotionality. In this test, " . . . a menu of paired lottery choices is structured so that the crossover point to the high-risk lottery can be used to infer the degree of risk aversion" (Holt and Laury 2002). The lottery consists of ten pairs of A and B options where the results depend on the cast of a ten-sided die. Option A awards either $20.00 or $16.00, while Option B awards $38.50 or $1.00. At first, the higher payoff is very unlikely, occurring only if the die result is 1, but it gradually increases by 1 until the last lotto, where the probability of a higher payoff is 10 out of 10. Riskier people might choose the less likely, but more rewarding option B, whereas conservative people might choose option A at first. By lotto 5, a rational, risk-free behavior would switch from option A to option B, because the odds of a higher payoff are 50%. However, while participants consistently opt for the safer A lotteries in pairs 1–3, they do not as consistently switch to

B lotteries around pair 5 (Holt and Laury 2002). The pair at which a person switches to B lottery is taken as a measure of the person's risk aversion: risk-seeking people switch to lottery B earlier, risk-neutral people switch around pair 5, risk averse people switch to lottery B later. In this study, we measured risk attitudes in both the participants' heritage language (Spanish) and their second but more dominant language (English). Previous research on FLe has shown that risk aversion is decreased in the foreign (and thus less emotional) language (Costa et al. 2014a), and we seek to establish whether such an effect is present in either of the heritage speakers' language.

### 2.4.2. Purely-Cognitive Tasks

These tasks included two tasks—an extended version of the Cognitive Reflection Test (CRT) (Frederick 2005) and the Disjunction fallacy (Bar-Hillel and Neter 1993).

Cognitive Reflection Test (CRT): This test consists of three logic problems with no emotional component where answering correctly requires participants to suppress answers that intuitively seem correct. Consider one of the problems of the test:

A bat and a ball together cost $1.10. The bat costs a dollar more than the ball. How much does the ball cost?

An intuitive answer is 10 cents, but the correct answer is 5 cents. Out of 3428 participants mentioned in Frederick (2005), 33% gave incorrect answers to all of the CRT problems, and 83% of the respondents missed at least one of the three questions, revealing lack of " . . . ability or disposition to reflect on a question and resist reporting the first response that comes to mind" (p. 35), and hence also revealing routine use of spontaneous, quick, and effortless heuristics. Importantly, if FLe is caused by decreased emotionality in one of the languages, there should be no effect of language on their performance on this test, since it does not involve an emotional component as would risk/loss aversion. We extended the task by adding 5 problems that tap into the same abilities as the original CRT. The final version of the task consisted of eight problems (See Appendix C).

Disjunction fallacy: Disjunction fallacy refers to people's tendency to estimate a disjunctive statement to be less probable than at least one of its component statements. For example, consider the following statement:

Mr. Pius goes to church every Sunday. He gets most of his information about religion from church and does not really read the Bible too much. Mr. Pius has a figurine of St. Mary at home. Last year, when he went to Rome, he toured the Vatican. From this information, Mr. Pius is more likely to be Catholic than Catholic or Muslim.

The answer to the statement is "false", because the probability of Mr. Pius being Catholic OR Muslim is higher than of him being just Catholic, because there are more Catholics and Muslims combined than there are just Catholics. Nonetheless, people tend to give a positive answer to this statement, succumbing to the disjunction fallacy. Similar to our predictions for the Cognitive Reflection Test, if FLe observed in the FL learners stems from reduced emotionality in the foreign language, we should not find an effect of language in our HS speakers; if it is caused by increased deliberation caused by the use of the less proficient FL, we should find bias reduction in their L1 (Spanish), since it is their less dominant and less proficient language; finally, if it is caused by a cognitive overload due to the use of the less proficient language, bias reduction should be present in their L2 (English).

The experiment was conducted online using the Qualtrics platform (Qualtrics, Provo, UT, USA). In order to avoid priming on similar tasks across languages, we randomly divided participants into four separate groups, varying task, frame of presentation and language. For example, some tasks were presented as a gain and as a loss in both English and Spanish, as seen in Table 4. Other tasks that did not involve two presentations were only divided by language.

**Table 4.** Study design.

| Language | Task | |
|---|---|---|
| | **Presentation 1** | **Presentation 2** |
| English | Group 1 | Group 2 |
| Spanish | Group 3 | Group 4 |

The paper excludes data from some non-HS participants who were originally included in the study, a fact that generated some asymmetries in the total number of participants per group. A summary of the total number of participants per group and task is given in Table 5.

**Table 5.** Summary of participants per task.

| Task | English | Spanish |
|---|---|---|
| Asian disease | 54 | 44 |
| Financial crisis | 44 | 54 |
| Ticket loss | 22 | 23 |
| Money loss | 32 | 22 |
| Lotto | 40 | 49 |
| Extended version of the CRT | | |
| (8 tasks) | 271 | 259 [1] |
| Disjunction fallacy | 44 | 44 |

[1] Participants responded to 4 CRT tasks in English and 4 in Spanish, but some participants left some of the tasks unanswered.

## 3. Results

In this section, we will first present results from decision-making tasks that involve an emotional component (Asian disease, Financial crisis, Ticket/Money loss, and Lotto), followed by results from tasks that are purely-cognitive (the Cognitive Reflection Test and Disjunction fallacy).

### 3.1. Cognitive-Emotional Decision-Making Tasks

3.1.1. Risky Choice Framing Tasks: Asian Disease and Financial Crisis Problems

Figure 2 represents combined percentages of safe responses (option A) in the gain frame vs. loss frame obtained in the Asian disease and Financial crisis problems. The x axis also represents the first and second language acquired by participants. As the figure shows, the results from our study reveal the following trends: the percentage of safe (option A) responses was higher in the gain than in the loss frame in both languages. Furthermore, English (the L2 in this case) showed a tendency to reduce framing bias.

We first fit a generalized linear model (R, lme4) with Decision (option "A" or option "B") as a dependent variable and Framing ("Gain" or "Loss") as predictor. The results in Table 6 indicate that Framing was indeed a significant factor in selecting option "A" (safe option) or "B" (gambling option), a result consistent with previous studies on framing effects. Specifically, the loss frame decreases the odds of choosing "A". In other words, people tend to choose the gambling option more often when the problems are presented as losses.

**Table 6.** Regression model, Asian disease and Financial crisis combined.

| Combined Asian + Financial | Est. | SE | *p* |
|---|---|---|---|
| Framing (Loss) | 0.85 | 0.30 | 0.004 |

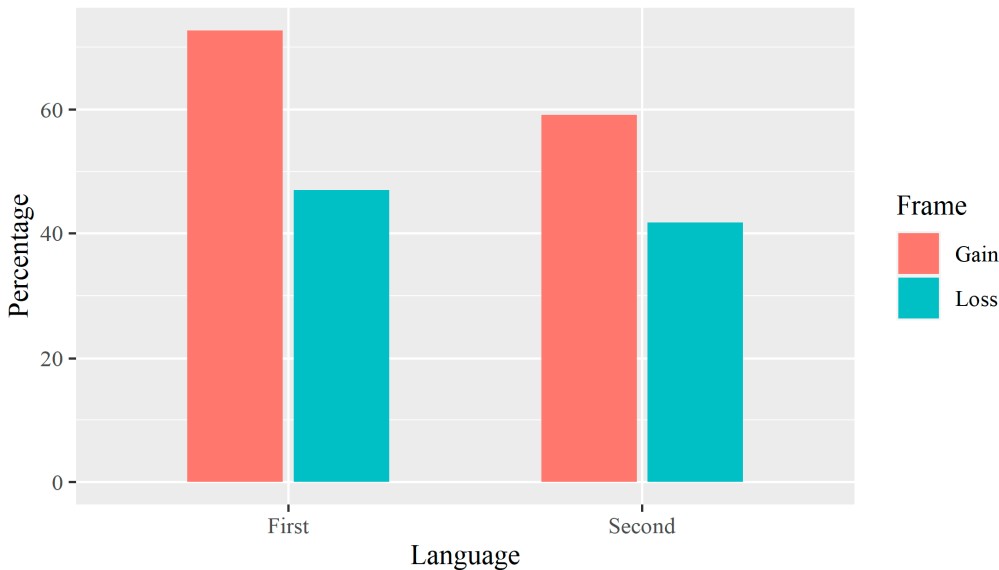

**Figure 2.** Combined percentage of safe ("A") responses for the Asian disease and Financial crisis problems in the first and second language.

In order to assess whether the language of presentation had an effect on decisions, a second model was fit with Decision (option "A" or option "B") as a dependent variable, and Framing ("Gain" or "Loss") and Language ("Spanish" or "English") as independent variables. As seen in Table 7, this model revealed a statistically significant effect for Framing, with the loss frame decreasing the odds of choosing "A", but it failed to reveal an effect for Language.

**Table 7.** Regression model, Asian disease and Financial crisis.

| Combined Asian + Financial | Est. | SE | *p* |
|---|---|---|---|
| Framing (Gain) | 0.9 | 0.30 | 0.004 |
| Language (Spanish) | 0.4 | 0.30 | 0.13 |

We also tested whether proficiency, as measured by the standardized test, had an effect on the subset of data that was presented only in Spanish. The model used "Framing" and "Proficiency" as predictors of "Decision". Only Framing had a significant effect on selecting "A" or "B", as seen in Table 8.

**Table 8.** Spanish items (Asian disease and Financial crisis combined).

| Combined Asian + Financial | Est. | SE | *p* |
|---|---|---|---|
| Framing (Loss) | −1.00 | 0.46 | 0.03 |
| Proficiency | −0.002 | 0.01 | 0.86 |

In conclusion, and consistent with the existing monolingual literature, we found a significant effect of framing in the two risky choice framing tasks: HS speakers in our study tended to choose gambling option B reliably more often in the loss framing. However, neither language of presentation nor proficiency in Spanish had an effect on the framing bias.

3.1.2. Mental Accounting of Economic Outcomes: Ticket/Money Loss Problem

In this task, participants were given two possible scenarios whose outcome would be identical. In the first one, the local scenario, tickets are lost and participants are asked if they would re-purchase them, whereas in the second one, the *global* scenario, money is

lost and participants are asked if they would spend additional money to purchase tickets. Results in Figure 3 show that participants were more likely to say "yes" to spending money in the money-expenditure (global) option than in the ticket-purchase (local) option both in English and Spanish. In addition, the percentage of "yes" responses in local and global scenarios was more similar in English (L2) than in Spanish.

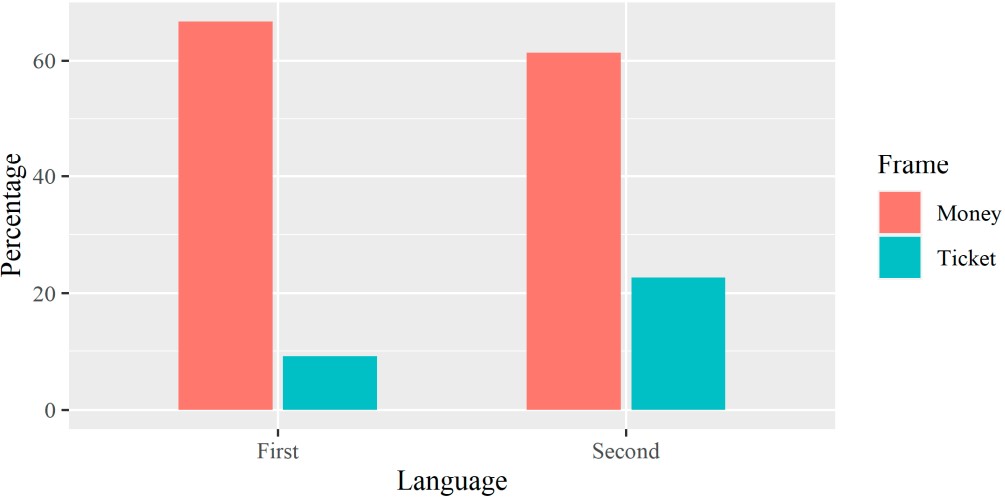

**Figure 3.** "Yes" responses in money/ticket task.

We fit generalized linear models using R (lme4) with Decision ("No" or "Yes") as dependent variable and Option ("Money" or "Ticket") and Language ("English" or "Spanish") as predictors. As reported in Table 9, the model showed a robust significant main effect of Option, but not of Language: presenting the problem as a *money* loss significantly increased the probability of deciding "Yes", but any differences across the two languages were not statistically significant.

**Table 9.** Regression results for Ticket/Money loss tasks.

| Combined Ticket + Money | Est. | SE | *p* |
|---|---|---|---|
| Option (Money) | −2.99 | 0.87 | 0.0006 |
| Language (English) | 1.07 | 0.47 | 0.23 |
| Option x Language | −1.31 | 1.07 | 0.22 |

In order to investigate the role of Spanish proficiency, we selected the subset of responses in Spanish and ran a separate model with Decision ("No" or "Yes") as a dependent variable and Proficiency and Option as predictors. Table 10 indicates that both of the variables had a marginally significant effect on the responses. Money had a positive effect on "yes" answers, while proficiency had a slightly negative effect, so higher proficiency resulted in slightly lower odds of selecting "yes".

**Table 10.** Proficiency and framing effects, Spanish task items only.

| Combined Ticket + Money | Est. | SE | *p* |
|---|---|---|---|
| Option (Money) | 2.42 | 0.92 | 0.008 |
| Proficiency | −0.07 | 0.04 | 0.06 |

In conclusion, presenting the problem as a money loss resulted in increased odds of choosing "yes" (willingness to repurchase tickets), while higher proficiency slightly decreased those odds.

### 3.1.3. Risk Aversion under Uncertainty: Hault-Laury Lottery

In the final cognitive-emotional task, participants were presented with 10 paired A and B lotteries and had to select either lottery A (more conservative option) or lottery B (gambling option). The blue "neutral" line in Figure 4 illustrates rational, risk-neutral behavior—choosing lottery A up until the pair 4 and switching to lottery B at pair 5. As red and green lines in Figure 4 illustrate, our participants revealed risk-aversive behavior in both of their languages: on average, 61.5% of them chose the safer option "A" in the lotto 5, although rational, risk-neutral behavior would be to choose option B at this crossover point.

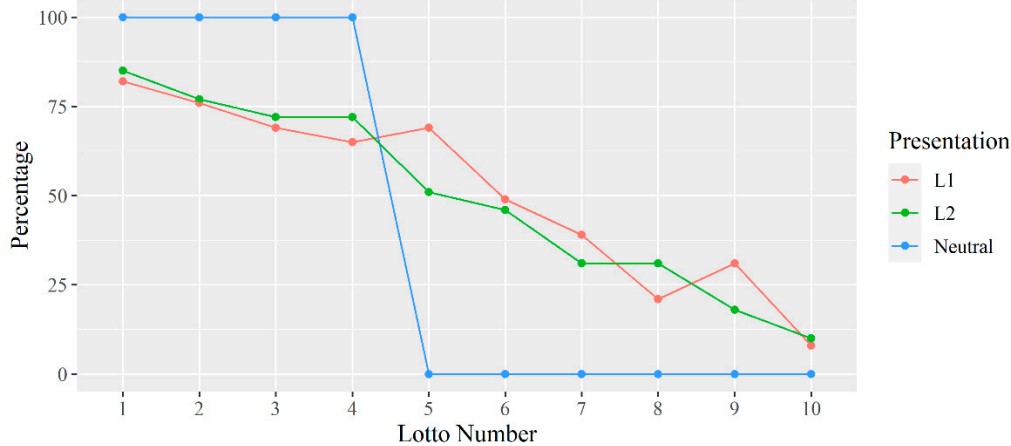

**Figure 4.** Percentage of "A" choices in the Lotto task.

Furthermore, the distribution of responses in Spanish vs. in English for the critical 5th lottery pair was not significantly different, as seen in Table 11, although "A" choices were selected more frequently in Spanish than in English (L2).

**Table 11.** Percentage of "A" choices in the Lotto task by language.

| Lotto Pair 5 | Est. | SE | $p$ |
|:---:|:---:|:---:|:---:|
| Language | 0.67 | 0.44 | 0.12 |

We also fit a model for the Spanish data to see if Spanish proficiency affected the choice for the 5th lottery pair, but it did not, as seen in Table 12.

**Table 12.** Proficiency effect on choice for lottery 5.

| Lotto Pair 5 | Est. | SE | $p$ |
|:---:|:---:|:---:|:---:|
| Proficiency | −0.01 | 0.01 | 0.39 |

### 3.2. Purely-Cognitive Tasks

Purely-cognitive tasks involve mental calculations but do not involve an emotional component. The two cognitive tasks used in this study are an extended version of the Cognitive Reflection Test (Frederick 2005) and Disjunction fallacy (Bar-Hillel and Neter 1993).

#### 3.2.1. Cognitive Reflection Test (CRT)

Results of the extended version of CRT are presented in Table 13. The left column represents the eight different tasks (Bat, Machine, etc.). Each of the tasks has a correct answer, an intuitive answer that is not correct, and an answer that is neither the correct one nor the intuitive one. For example, for the Egg task, the question is "It takes 5 min to boil an egg. How much time does it take to boil 4 eggs?". The correct answer is 5 min (the time to boil 1 egg vs. 4 eggs is the same), the intuitive answer is 20 min, and an example

of "other" answer would be a random number like 14 min. We present results for each individual task, coded either as "correct", "intuitive", or "other".

**Table 13.** Cognitive Reflection Test results (total count and percentages per answer type per language for each task).

| | **English** | | | | **Spanish** | | | |
|---|---|---|---|---|---|---|---|---|
| | **Intuitive** | **Correct** | **Other** | | **Intuitive** | **Correct** | **Other** | |
| **Tasks** | | | | **Total** | | | | **Total** |
| | **Answer** | **Answer** | **Answer** | | **Answer** | **Answer** | **Answer** | |
| Bat | 21 | 13 | 2 | 36 | 19 | 7 | 5 | 31 |
| (%) | 58 | 36 | 6 | 100 | 61.5 | 22.5 | 16 | 100 |
| Machine | 24 | 9 | 6 | 39 | 20 | 7 | 1 | 27 |
| (%) | 61.5 | 23 | 15.5 | 100% | 71.5 | 25 | 3.5 | 100 |
| Division | 16 | 11 | 2 | 29 | 19 | 14 | 3 | 36 |
| (%) | 55 | 38 | 7 | 100 | 53 | 39 | 8 | 100 |
| Airplane | 12 | 15 | 1 | 28 | 26 | 12 | 3 | 41 |
| (%) | 43 | 53.5 | 3.5 | 100 | 63.5 | 29 | 7.5 | 100 |
| Doctor | 29 | 9 | 0 | 38 | 19 | 6 | 2 | 27 |
| (%) | 76 | 24 | 0 | 100 | 70.5 | 22 | 7.5 | 100 |
| Egg | 13 | 19 | 3 | 35 | 15 | 17 | 3 | 35 |
| (%) | 37 | 54.5 | 8.5 | 100 | 43 | 48.5 | 8.5 | 100 |
| Ribbon | 11 | 18 | 6 | 35 | 12 | 13 | 4 | 29 |
| (%) | 31.5 | 51.5 | 17 | 100 | 41 | 45 | 14 | 100 |
| Lilies | 16 | 10 | 3 | 29 | 15 | 14 | 5 | 34 |
| (%) | 55 | 34.5 | 10.5 | 100 | 44 | 41 | 15 | 100 |
| Total | 142 | 104 | 23 | 269 | 145 | 90 | 26 | 260 |
| (%) | 53 | 39 | 8 | 100 | 55 | 35 | 10 | 100 |

A multinomial model (R, nnet) was fit for the combined tasks, with Answer type ("intuitive," "correct," or "other," where "correct" serves as reference level) as outcome variable and Language ("English" or "Spanish") as predictor. When moving from Spanish to English, the odds of an "intuitive" answer decreased by 0.85 compared to a "correct" answer, as did the odds of an "other" answer compared to a "correct" one (0.77), however these effects were not statistically significant.

We ran a second model with the Spanish subset of data, with Answer Type ("intuitive," "correct," or "other") as outcome variable and "Proficiency" as predictor. Increasing one unit in proficiency decreased the odds of selecting an "intuitive" answer only minimally (0.99), suggesting that proficiency had a negligible effect. The odds of selecting "Other" answers were reduced as proficiency increased, but also insignificantly so (0.93).

To sum up, in the CRT task, Language of Presentation and Proficiency had an insignificant effect on HS speakers' responses.

### 3.2.2. Disjunction Fallacy

The results of the task are presented in Table 14. As in the original Disjunction fallacy studies (Carlson and Yates 1989; Bar-Hillel and Neter 1993), more people chose the "True" option (which violates probability laws and is thus considered incorrect) in both languages. For example, when presented with the task in English, out of 95 total responses, 69 were "true" responses (73%) and 26 were "false" responses (27%), with a difference of 46% between true and false responses. For Spanish, this difference was even more prominent (68%), with the incorrect responses totaling up to 84%.

**Table 14.** Disjunction fallacy results (total number of true and false responses and percentages per language).

| Disjunction Fallacy | English | | | Spanish | | |
|---|---|---|---|---|---|---|
| | **True** | **False** | **Diff. T-F** | **True** | **False** | **Diff. T-F** |
| | 69 (73%) | 26 (27%) | 46% | 81 (84%) | 15 (16%) | 68% |
| Total | | 95 | | | 96 | |

We ran a generalized linear model (R, lme4) with Decision ("True" or "False") as dependent variable and Language ("English" or "Spanish") as predictor. Results in Table 15 indicate that the language of presentation significantly affects choice of "True" or "False": the likelihood of selecting the correct "False" response decreased as the presentation shifted from English to Spanish. In other words, heritage speakers were significantly more likely to provide the correct "False" response in their second, more proficient, language (English).

**Table 15.** Disjunction fallacy regression by language.

| Disjunction Fallacy | Est. | SE | *p* |
|---|---|---|---|
| Language (English) | −0.76 | 0.36 | 0.04 |

For the subset of items presented in Spanish, proficiency was also statistically significant, as presented in Table 16: the probability of selecting the correct "False" response increased with proficiency.

**Table 16.** A Disjunction fallacy regression, Spanish items by proficiency.

| Disjunction Fallacy | Est. | SE | *p* |
|---|---|---|---|
| Proficiency | 0.4 | 0.20 | 0.04 |

In sum, our results confirm results from previous monolingual studies regarding presentation biases: we found an overall effect of presentation across the tasks. In addition, we found a statistically significant effect of language and proficiency in one of our purely-cognitive tasks, the Disjunction fallacy task, where participants chose the appropriate "False" response more frequently in English and also as their proficiency in Spanish increased.

Table 17 presents a summary of results. Across most tasks, English reduced the bias, although not significantly so, with the exception of the Disjunction fallacy, where the effect was significant.

**Table 17.** Results summary.

| | Overall Bias | Reduction of Bias Depending on Language | Effect of Proficiency in Spanish |
|---|---|---|---|
| Asian + Financial crisis | Yes (*) | English (L2, n.s) | n.s |
| Money/Ticket loss | Yes (*) | English (L2, n.s) | n.s |
| Lotto (Hault-Laury) | Riskier (n/a) | English, less risk-averse (n.s) | n.s |
| CRT | Yes (n/a) | English (L2) | Yes |
| Disjunction fallacy | Yes (n/a) | English (L2, *) | Yes (*) |

* Statistically significant. n.s Not statistically significant. n/a Not applicable.

## 4. Discussion

This study contributes to the discussion of FLe by providing data from the novel and principally distinct population of Spanish heritage speakers (HS speakers). FLe

refers to an observed bias reduction in decision-making tasks administered in a foreign language. As noted earlier, initial studies with FL learners found FLe in decision-making tasks and moral dilemmas; this FLe has been accounted for by a reduced emotionality in the FL. More recent studies have engaged populations of non-FL speakers such as highly proficient acculturated bilinguals to determine whether FLe is (1) generalizable to a different population of language speakers, and (2) caused by reduced emotionality in one of the languages. Our study extends this research program to a unique population of L1 Spanish/L2 English HS speakers using the standard battery of judgment and decision-making tasks. In the next few paragraphs, we discuss results from EPT (Emotional Phrases task) first, followed by the findings on cognitive biases in general, and finally findings on FLe.

First, we replicated and confirmed the findings of emotionality studies (Anooshian and Hertel 1994; Harris et al. 2003; Harris et al. 2006; Sutton et al. 2007; Eilola and Havelka 2011; Ferré Pilar et al. 2010; Caldwell-Harris et al. 2011; Ferré et al. 2018; Ivaz et al. 2019; Miozzo et al. 2020) which conclude that an early bilingual has comparable emotional resonances in both languages. We tested HS speakers on the Emotional Phrases Task, which has been used in previous research on emotionality and language (Harris et al. 2003; Harris 2004; Caldwell-Harris and Ayçiçegi-Dinn 2009). As expected, and in line with the previous studies, our HS speakers did not overall rate the EPT phrases higher in their L1 than in the L2: there were no differences between the Spanish and English ratings for 20 phrases out of the total 22.

We acknowledge that emotionality has different dimensions that may not be immediately captured by behavioral tasks such as the one used in this study. Many studies on emotionality and language supplemented emotionality ratings with psychophysiological measures such as skin conductance tests (Harris et al. 2003; Harris 2004; Eilola and Havelka 2011). In fact, some report a dissociation between the findings from behavioral measures vs. psychophysiological ones. For example, Harris et al. (2003) found no differences between the L1 Turkish and L2 English ratings of emotional words, but they did find significantly stronger skin conductance responses in Turkish. In short, EPT may be informative, but it certainly is neither exhaustive nor fully reliable in all cases.

Despite these considerations, our EPT results are consistent with 1) data from a significant number of research studies that show no differences between emotional resonances in the two languages of early bilinguals, and 2) with the general consensus among scholars that there is a comparable emotional intensity of the two languages spoken by an early bilingual. For these reasons, we may be able to conclude that our HS speakers have similar emotional reactions to their two languages in general. Beyond this broad claim, there could be a myriad of variables that affect emotionality. For example, different linguistic stimuli may trigger variable degrees of emotionality in the two languages depending on the language in which the stimuli were first and/or mostly experienced. For example, childhood reprimands would potentially be experienced by HS speakers more emotionally in the L1, whereas romantic endearments would be so in the L2 because these are the languages in which they were most likely to encounter them. This idea is supported by studies showing that autobiographical memories are recalled more easily in the language in which they were experienced (Marian and Neisser 2000). Overall, emotionality is a highly complex concept and, as McFarlane and Cipolletti Perez (2020) argue, one that is hard to measure in the absence of a predictive and generalizable theory of emotion and specific emotion types.

Second, as far as our findings on cognitive biases in judgment and decision-making (JDM) are concerned, they are consistent with previous monolingual literature: HS speakers clearly exhibit cognitive biases across all tasks in both languages. Specifically, in cognitive-emotional tasks they fall victim to framing biases, show risk aversion, and have a preference for local vs. global accounting. In purely cognitive tasks, they tend to provide intuitive rather than rational answers on CRT and provide answers that violate probability laws in the Disjunction fallacy task. Thus, this study is the first one to demonstrate cognitive biases

in heritage speakers. While there is no reason to suspect that cognitive biases do not apply to heritage speakers, this was an empirical question that we have addressed in this paper.

Third, as far as FLe is concerned, we found a general tendency toward bias reduction in the L2; however, this tendency was not statistically significant in cognitive-emotional tasks, a finding that differs from some previous studies on FL effects. For purely-cognitive tasks (CRT and Disjunction fallacy) the results were more complex: language effect was not significant for the CRT, but it was significant for the Disjunction fallacy. More specifically, CRT correct responses were more likely to be given in English, with the increase in correct responses corresponding to a decrease in "other" responses, rather than to a decrease in intuitive ones. It is possible that participants are more used to doing mathematical calculations in English, since for most of them, this would have been the language of their schooling beginning at an early age. Crucially, we did find a significant effect of language of presentation on Disjunction fallacy responses—there was a statistically significant reduction of incorrect (intuitive) responses in the L2 compared to the L1. In other words, HS speakers answered the Disjunction fallacy problem more correctly in their second, more proficient language. Moreover, there was a significant effect of proficiency in Spanish on the responses, such that higher proficiency was correlated with more correct responses when the task was presented in Spanish.

We will now discuss these results in light of each of the three hypotheses proposed in this paper. The predictions from Table 2 are partially reproduced in Table 18.

**Table 18.** Predictions from the hypotheses.

| Hypothesis | Predictions for Heritage Speakers |
|---|---|
| Reduced emotionality | If both equally emotional, then bias-reduction in neither language in any of the tasks |
| Cognitive enhancement | Bias-reduction in less proficient L1 across all tasks |
| Cognitive overload | Bias-reduction in more proficient language across all tasks |

Our findings are not fully compatible with the reduced emotionality hypothesis for two reasons. First, if the cause of FLe on JDM were reduced emotionality in the L2, we should essentially find no differences at all in our HS population, given that their two languages are comparable in terms of emotionality. Second, even if our EPT is incorrect and our HS speakers do have unequal emotionality in one of their languages, the FLe should be present only in tasks that involve emotionality—the cognitive-emotional tasks. Although we found a tendency for English to reduce biases, we did not find any significant differences between the two languages as far as JDM in cognitive-emotional tasks was concerned, and we found a significant effect of language of presentation and proficiency in one of our purely-cognitive tasks (Disjunction fallacy). Nonetheless, we are cautious about making strong claims with respect to the reduced emotionality hypothesis, because our HS speakers are presumably equally emotional in both languages, so the null effect could both serve as evidence against the hypothesis as well as a mere indication that we do not have enough participants to find an effect of emotion.

Our findings are also not compatible with the cognitive enhancement hypothesis, since it predicts a bias reduction in the less proficient language, and our results show the opposite effect. Similarly, it predicts that higher proficiency will lead to increased biases, which runs contrary to our findings: lower proficiency in Spanish led to significantly lower scores in the Disjunction fallacy.

Finally, our results are most consistent with the cognitive overload hypothesis. Recall that this hypothesis states that the ability to provide a rational response should be affected in the less proficient FL, and that researchers hypothesized that these reduced rational responses were due to the increased processing load in that language. Since English is the dominant and more proficient language for our heritage speakers, we would expect fewer cognitive-load effects in that language. While we did not find significant differences in most tasks, we did find them in one of the tasks, where the HS speakers responded overall

more correctly in their more proficient language (L2 English). Furthermore, in the less proficient language (L1 Spanish), those with higher proficiency responded more correctly than those with lower proficiency.

One might object that if the cognitive overload hypothesis were correct, we would find language of presentation effects in more tasks, not just one of them, and that proficiency would have a larger effect on bias reduction. We acknowledge that finding a stronger effect across more tasks would make a more robust case for the cognitive overload hypothesis, and we suspect that we did not find it for several reasons. First, our HS speakers may not have low enough proficiency to suffer sufficient overload to affect their responses. Studies have shown that FL learners are affected by processing load (Meuter and Allport 1999), but that early balanced bilinguals are not (Costa and Santesteban 2004; Costa et al. 2006). Although our HS speakers are early bilinguals and are more proficient than FL learners, they are still not balanced in both languages; hence, they might represent an intermediate case between FL learners and early balanced bilinguals, where processing load still affects them, but to a lesser extent than FL learners. Moreover, our HS speakers' proficiency fell on a continuum from very high to very low and it certainly could have introduced variability in degrees of cognitive load. Having proficiency groups with consistently low proficiency (but sufficient to understand the tasks) might have revealed significant effects across more tasks. This is an empirical question that future research can investigate. In addition, as an anonymous reviewer suggests, the foundational methodologies of this study could be strengthened through a rigorous proficiency measure of the L2 (English). Although we assumed that our participants were native speakers—since they were exposed to English in early childhood and all of their schooling was in English, including higher education—and we thus expected them to perform at ceiling on English proficiency tests, future studies of this nature might benefit from incorporating an additional linguistic measure of the heritage speakers' L2 skills beyond self-ratings.

Another possible interpretation of these results is that language effect does not apply at all to heritage speakers: if it can be shown that this is due to their being equally emotional in both languages, it would provide support for the reduced emotionality hypothesis, and this is the reason why we do not claim to have refuted it. We also acknowledge that the statistical power of this study is lower than some of the previous ones. For a medium size effect, 47 participants per language results in .73 statistical power. Finally, it is possible that reduced emotionality, cognitive enhancement, and cognitive overload operate simultaneously, producing a combined language effect. In any case, we should assume that any language effect should be as unique as the individual bilingual speaker, and should depend on their specific linguistic profile (language proficiency, dominance, emotionality, etc.).

As such, even the issue of whether FLe exists at all remains unresolved: some studies using FL populations have shown that foreign language decreases cognitive biases and increases benefit-maximizing inclinations (Keysar et al. 2012; Costa et al. 2014a; Geipel et al. 2016; Dylman and Champoux-Larsson 2020), while others show the opposite (Hayakawa et al. 2017; Muda et al. 2018; Białek et al. 2019; Mills and Nicoladis 2020); some studies using highly proficient, acculturated bilinguals have found FLe (Brouwer 2020; Miozzo et al. 2020), while others have not (Čavar and Tytus 2018; Brouwer 2019; Dylman and Champoux-Larsson 2020).

Nevertheless, the contribution of the present study to the FLe research is tangible: first, we have explored language effects on decision-making in a novel bilingual population of heritage speakers; second, we addressed the issue of confounding emotionality and proficiency inevitable in FL learners; and, finally, we showed that if a language effect exists in HS speakers, who have equal emotional resonances in both Spanish and English, it would be caused by cognitive overload in the less proficient language rather than by cognitive enhancement. This is consistent with recent moral decision-making research suggesting that low proficiency in a language is not correlated with heightened utilitarianism (Hayakawa et al. 2017; Muda et al. 2018; Białek et al. 2019).

Lastly, we would like to suggest potential avenues for future research on FLe. One future direction is the direct measurement of cognitive load and emotionality during the process of decision-making. While cognitive load and emotionality have been suggested as key factors causing FLe on decision-making tasks, no studies, to the best of our knowledge, have tested the amount of cognitive load or emotional reaction *during* the decision-making tasks. It would be enlightening to measure both the amount of cognitive load as well as the intensity of emotional response induced during the L1 vs. FL presentations of the task: if cognitive load is reliably higher in the FL condition than in the L1 condition, one can conclude that FLe is related to the amount of cognitive load induced by the FL. Some objective measures of cognitive load include pupillometry (eye-tracking), brain activity measures such as MRI and fNIRS, EEG and cardiovascular metrics; while subjective measures include self-reports of stress or mental effort (Martin 2014). Similarly, it would be highly informative to measure emotionality level during the task. While we did employ a measure of emotionality (Emotional Phrases Task) to get insights into which of the two languages overall is perceived as more emotional by our HS speakers, we did not measure emotionality induced by the task itself. As mentioned before, some studies on language and emotionality used skin-conductance tests (Harris et al. 2003; Harris 2004; Eilola and Havelka 2011), but none of them studied the FLe as it specifically impacts decision-making. FLe research would also benefit from extending it to the other types of heuristics, such as fast and frugal heuristics (Chen et al. 2015). This is because most research on FLe to date, including ours, is based on Tversky and Kahneman's program, which considers heuristics to be a liability rather than a tool, but this assumption represents just one of many views on heuristics within the philosophy of judgment and decision-making.

**Author Contributions:** Conceptualization, A.K. 50% and J.C. 50%; methodology, A.K. 50%, J.C. 50%; statistical analysis, J.C. 100%; writing—A.K. 65%, J.C. 35%. All authors have read and agreed to the published version of the manuscript.

**Funding:** This research received no external funding.

**Institutional Review Board Statement:** The study was conducted according to the guidelines of the Declaration of Helsinki, and approved by the Institutional Review Board (or Ethics Committee) of Rutgers University (protocol code E15-333 approved on 1 January 15).

**Informed Consent Statement:** Informed consent was obtained from all subjects involved in the study.

**Data Availability Statement:** The data presented in this study are available on request from the corresponding author. The data are not publicly available due to privacy restrictions encoded in the original IRB protocol.

**Acknowledgments:** We would like to acknowledge Dina Rizk, Amanda Figueroa and Michelle Villavicencio, who worked on this project as research assistants, and John Sarkissian and Steven Reale, who helped us with editing and proofreading. We also thank the anonymous reviewers who generously de-voted time and made thoughtful comments on previous versions.

**Conflicts of Interest:** The authors declare no conflict of interest.

## Appendix A

**Sociolinguistic background questionnaire**

What is your age?
Country of origin
How long have you lived in the United States?

**Spanish English is spoken**

Number of years and months you spent in a country where
Number of years you spent in a family where
Number of years you spent in a school where
Number of years and months you spent in a working environment where

List all the languages you know in decreasing order by how well you know them. Put the one you know best at the top

List all the languages you know in the order in which you learned them (your native language(s) should go first)

At what age (in years) did you start to learn each of the following languages
  English    Spanish    Other (please specify)

Please indicate, on average, the percentage of the time you were exposed to the following languages outside the home between 0–5 years (percentages should add up to 100%).
  English    Spanish    Other (please specify)

Please indicate the overall percentage of time you are currently and on average exposed to each of the following languages (percentages should add up to 100%)
  English    Spanish    Other (please specify)

What is your parents' native language (mark both if they are bilingual from birth)

|  | English | Spanish | Other (please specify) |
|---|---|---|---|
| Father |  |  |  |
| Mother |  |  |  |
| Other family members in household |  |  |  |

In which years were you educated in each of your languages and each of these areas?

|  | Language 1 | Language 2 | Language 3 |
|---|---|---|---|
| Your overall education |  |  |  |
| Reading and writing |  |  |  |

How many years of formal education have you had
  In the US    In another country (specify the country)

When choosing a language to speak with a person who is equally fluent in all your languages, what percentage of time would you choose to speak each language? Please report percent of total time. (percentages should add up to 100%)
  English    Spanish    English and Spanish (in the same conversation)    Other

Please rate your abilities (0 = poor, 5 = native speaker command) in each of the following areas in

|  | English | Spanish |
|---|---|---|
| Overall |  |  |
| Reading |  |  |
| Writing |  |  |
| Listening |  |  |
| Speaking |  |  |

**Appendix B**

| Sentence | English | Spanish |
|---|---|---|
| 22 | The chairs in this lounge are comfortable. | Las sillas de esta sala son muy cómodas. |
| 21 | White shoes get easily dirty. | Los zapatos blancos se ensucian fácilmente. |
| 20 | The ladder is leaning against the wall. | La escalera está apoyada en la pared. |
| 19 | Squirrels jump from tree to tree. | Las ardillas saltan de árbol en árbol. |
| 18 | I can't wait to see you! | ¡Tengo muchas ganas de verte! |
| 17 | I was working in the afternoon. | Estaba trabajando por la tarde. |
| 16 | I saw you at the store yesterday. | Te vi en la tienda ayer. |
| 15 | I am sick of you! | ¡No te soporto más! |
| 14 | You are everything to me! | ¡Eres mi vida! |
| 13 | I hate you! | ¡Te odio! |
| 12 | The book is on the table. | El libro está sobre la mesa. |
| 11 | You are so stupid! | ¡Eres tan estúpido/a! |
| 10 | The house is painted white | La casa está pintada de blanco. |
| 9 | You are so fat! | ¡Estás tan gordo/a! |
| 8 | I never want to see you again! | ¡No quiero volver a verte nunca! |
| 7 | Cyclists are generally skillful. | Los ciclistas normalmente son muy habilidosos. |
| 6 | I've missed you so much! | ¡Me has hecho mucha falta! |
| 5 | You are so ugly! | ¡Qué horrible eres! |
| 4 | I woke up at 9 this morning. | Me levanté a las 9 esta mañana. |
| 3 | The keyboard has numbers at the top. | El teclado tiene los números arriba. |
| 2 | I love you more than anything! | ¡Te quiero muchísimo! |
| 1 | I don't want to lose you! | ¡No quiero perderte! |

**Appendix C**

Extended version of the Cognitive reflection test

1. A bat and a ball together cost $1.10. The bat costs a dollar more than the ball. How much does the ball cost?
2. It takes 5 machines 5 min to make 5 keyboards. How long will it take 100 machines to make 100 keyboards?
3. In a lake there is a patch of lily pads. The patch doubles in size every day. If it takes 48 days for the patch to cover the entire lake, how long would it take for the patch to cover half the lake?
4. Divide 30 by $\frac{1}{2}$ and add 10. What is the answer?
5. An airplane travelling at 400 mph crashes on the US/Canadian border. Where are the survivors buried?
6. If it takes 20 min to hard-boil one goose egg, how long would it take to hard-boil 4?
7. A Doctor gives you three (3) pills and tells you to take one every half an hour. How long will it be until you no longer have any pills?
8. You have a ribbon that is 30 inches long. How many cuts with a pair of scissors would it take to divide it into inch long pieces?

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
