# Peer review of "Can You Make Better Decisions If You Are Bilingual?"

_languages, doi:10.3390/languages6010043_

Round 1

Reviewer 1 Report

I think the present paper offers a very interesting and insightful study dealing with how bilingualism affects decision-making. It examines decision-making skills and opportunities from the perspectives of emotion and cognition. I consider this a avenue of research worth further investigation. The languages we speak and how we relate to them emotionally and cognitivelly determine not only our thoughts, but also ur identity(ies). To me this is a great paper looking into these very relevant issues.   

Author Response

Thank you for your comments.

Reviewer 2 Report

This manuscript presents a study testing heritage speakers of Spanish on the foreign language effect. The study is original in that it is including bilinguals who are more proficient in their L2 English than in their L1 Spanish. The manuscript is well-written and easy to follow. I have a few major issues that could be addressed in revisions :

  • What is the difference between a purely cognitive task and a cognitive-emotional task? How would future researchers be able to tell how to classify a task appropriately?
  • Acknowledge some alternative explanations and/or limitations in the discussion. Notably, one possibly interpretation of their results is that the FLe doesn’t exist at all (for the one task that did show a significant difference by language, the N was quite low, so that result could be spurious). Some researchers (e.g., Chan et al. 2016) have found that the type of moral dilemma makes a huge difference in people’s responses.
  • A similar point to #2, if the authors are correct that processing load is the key variable, then future studies should include direct measures of processing load. Do they have any suggestions for what such a measure could be?

Smaller comments :

  1. 4, end of second paragraph : English is the L3, no?
  2. 6, second paragraph : here it is not clear why it is important that the participants were ‘matched’. If I have understood correctly, it is because participants were assigned to a particular task and presentation so they wanted to match participants across conditions. Is that correct?
  3. 6, over halfway through third paragraph : for the 4.8 in English and 4.15 in Spanish, what was the possible range on the scale?
  4. 6, third paragraph, last sentence : the authors have not shown that the English and the Spanish proficiency task are comparable so they cannot use this measure to claim that they are less proficient in Spanish. The self-ratings can be used to back up this claim.
  5. 6, last paragraph : Table 3 and this paragraph were not clear at all, at this point in the text. Putting them after the description of the tasks would make more sense.
  6. 7, toward the end : I think I was missing a critical bit of information to understand the dependent variable here. From the presentation, I thought that in a gain frame, participants should choose A over C and in loss frame, they should choose D over B. What am I missing?
  7. 11-15 : the values in the tables with either only one line or two lines could be reported in the text.
  8. 12, last paragraph : how does the model show an almost significant effect for Option? P = .0006 in Table 8. By convention, that sounds significant.
  9. 13, first paragraph : I cannot match up the sentences in this paragraph with the results in Table 9. It looks like proficiency is almost significant, too.
  10. 13, third paragraph : these participants may have been a tad more risk-adverse, but the difference looks fairly minor
  11. 13, Figure 4 : what is neutral?
  12. 13, title of Figure 4 : maybe put the words in the same order as the graphs?
  13. 16, second paragraph : I did not understand the relevance of the claim that Spanish speakers tend to rate closer to the central part of the scale for linguistic acceptability tasks. Consider deleting.
  14. 17, first to second lines : because ‘significant’ has a technical meaning, consider replacing with maybe ‘large’?

Reference

Chan, Y.-L., X. Gu, J. C.-K. Ng, and C.-S. Tse. 2016. “Effects of Dilemma Type, Language, and Emotion Arousal on Utilitarian vs Deontological Choice to Moral Dilemmas in Chinese–English Bilinguals.” Asian Journal of Social Psychology 19: 55–65.

Author Response

Thank you for your detailed comments. Below are the responses.

REVIEWER 2

This manuscript presents a study testing heritage speakers of Spanish on the foreign language effect. The study is original in that it is including bilinguals who are more proficient in their L2 English than in their L1 Spanish. The manuscript is well-written and easy to follow. I have a few major issues that could be addressed in revisions :

  • What is the difference between a purely cognitive task and a cognitive-emotional task? How would future researchers be able to tell how to classify a task appropriately?

Any decision-making task involves cognitive processing. However, while some decision-making tasks may trigger negative affect (unpleasant emotions such as anxiety, fear, shame, guilt, irritability, etc.) (Watson, Clark, & Tellegen, 1988), other tasks do not produce such an effect on people. We refer to the former tasks as cognitive-emotional tasks (tasks involving risk aversion and loss aversion, moral dilemmas, etc.), while to the latter – as purely cognitive tasks (e.g., the outcome bias, the conjunction fallacy, disjunction fallacy, base-rate neglect fallacy, the cognitive reflection test, etc.).  this comment has been addressed in lines 256-264 of the revised manuscript (Note that all line numbers refer to document with markup turned off).

  • Acknowledge some alternative explanations and/or limitations in the discussion. Notably, one possibly interpretation of their results is that the FLe doesn’t exist at all (for the one task that did show a significant difference by language, the N was quite low, so that result could be spurious). Some researchers (e.g., Chan et al. 2016) have found that the type of moral dilemma makes a huge difference in people’s responses.

We agree with this comment, and we address the issue in lines 862-869.

  • A similar point to #2, if the authors are correct that processing load is the key variable, then future studies should include direct measures of processing load. Do they have any suggestions for what such a measure could be?

We agree with this comment, and we address it in lines 879-891. We have to add, however, that manipulating and/or measuring cognitive load online is problematic and might yield a null result.

Smaller comments :

  1. 4, end of second paragraph : English is the L3, no?

Clarified in lines 189-197.

  1. 6, second paragraph : here it is not clear why it is important that the participants were ‘matched’. If I have understood correctly, it is because participants were assigned to a particular task and presentation so they wanted to match participants across conditions. Is that correct?

Sentence should not have been there and so has been removed.

  1. 6, over halfway through third paragraph : for the 4.8 in English and 4.15 in Spanish, what was the possible range on the scale?

Five possible points; we added that in line 312.

  1. 6, third paragraph, last sentence : the authors have not shown that the English and the Spanish proficiency task are comparable so they cannot use this measure to claim that they are less proficient in Spanish. The self-ratings can be used to back up this claim.

We rephrased this by saying “low” instead of using a comparative adjective “lower”. See line 326.  

  1. 6, last paragraph : Table 3 and this paragraph were not clear at all, at this point in the text. Putting them after the description of the tasks would make more sense.

We agree completely, and we have moved this part below.

  1. 7, toward the end : I think I was missing a critical bit of information to understand the dependent variable here. From the presentation, I thought that in a gain frame, participants should choose A over C and in loss frame, they should choose D over B. What am I missing?

The paragraph was problematic. We have rewritten it, and it now appears in lines 403-406.

  1. 11-15 : the values in the tables with either only one line or two lines could be reported in the text.

For reasons of readability, we think it is better to keep the values in the tables.

  1. 12, last paragraph : how does the model show an almost significant effect for Option? P = .0006 in Table 8. By convention, that sounds significant.

This was a mistake, and we have rectified it in the revised manuscript.

  1. 13, first paragraph : I cannot match up the sentences in this paragraph with the results in Table 9. It looks like proficiency is almost significant, too.

This was also a mistake, and we have rectified it in the revised manuscript.  

  1. 13, third paragraph : these participants may have been a tad more risk-adverse, but the difference looks fairly minor

Correct. This difference was not significant.

  1. 13, Figure 4 : what is neutral?

We have added a clarification in the text in lines 627-631.

  1. 13, title of Figure 4 : maybe put the words in the same order as the graphs?

We have changed figure 4, excluding Costa et al. The title has been changed.

  1. 16, second paragraph : I did not understand the relevance of the claim that Spanish speakers tend to rate closer to the central part of the scale for linguistic acceptability tasks. Consider deleting.

We have deleted this part.

  1. 17, first to second lines : because ‘significant’ has a technical meaning, consider replacing with maybe ‘large’?

We agree, and replaced it as suggested.

Reviewer 3 Report

This study investigates different hypothesis that attempt to explain the Foreign Language effect (FLe) found for framing biases. The authors test the reduced emotionality hypothesis, the cognitive enhancement hypothesis, and the cognitive overload hypothesis by asking heritage speakers (HS) to perform several decision making tasks. The rationale is HS experience high emotionality in both their first (the heritage language) and second languages, but that they are more proficient in their second language (the one they mostly use and that is the main language of the society they live in) than in their first language (which they use less extensively and mostly in private settings). Given that most FLe studies use participants who are presumably less emotional and less proficient in their second language, and both more emotional and proficient in their first language, it is impossible to tease apart the influence of proficiency and emotion in such samples. By using HS in combination with both emotional and non-emotional tasks, the authors have the possibility to investigate the effects of emotion and cognitive demands individually.

The manuscript addresses a relevant and important current issue in FLe research. It initially shows a lot of potential, and I thought that the choice of HS speakers in combination with emotional and non-emotional tasks to test the different hypothesis was a clever methodological choice. However, there are some major issues in the design and analyses that makes me hesitant to agree with the conclusions that are drawn by the authors. 

Introduction

The introduction is generally well written and presents the arguments clearly. However, there are discrepancies that must be addressed. I have issues mainly with the third hypothesis, the cognitive overload hypothesis, which draws parallels from the FLe on moral decision-making. It is unclear to me how it is related to the reduced framing effect. I believe that the introduction would present a stronger and more coherent argument if it focused on the first two hypothesis, or if the authors could explain and justify their hypothesis better. More specifically:

48-53: the authors refer to a study by Costa et al., 2014 where they suggest that lower emotionality leads to more utilitarian effects. However, this study investigated moral decision making, not the framing bias. These are two different decision making tasks that share common mechanism but where important differences are found. The authors need to present evidence that shows that framing bias tasks and moral decision making tasks are equivalent.

75-76: the authors use the term “utilitarian” and “more rational” as synonyms, but that is not really true. Utilitarian refers to the greater good of the many rather than the few, which is not necessarily more rational. In moral decision making tasks, such as the Footbridge dilemma, it is indeed more rational to sacrifice the life of one person to save the life of five persons, but the terms are not synonyms.

75-93: Here again the authors present evidence related to moral decision making. Furthermore, the study by Greene et al. 2008 investigate moral decision making in one language and does not address the FLe. It appears to me that this theoretical point of view, as presented by Hayakawa et al. 2017 in relation to the FLe, is related to the first two hypothesis. Indeed, they write that what they find suggests that the FLe in moral decision making is likely due to reduced emotionality rather than increased deliberation. Indeed, the authors of this manuscript write “That is, if there is a FLe, it is actually the opposite of what has been assumed in the earlier studies - an increased cognitive load does not enhance but rather affects one’s ability to provide a utilitarian answer”. However, this is not what the FLe in moral decision making finds, is it? Usually, it finds that utilitarian responses increase when using a second language, not diminish. Furthermore, this is related to moral decision making, not framing bias. Could the authors clarify this point in relation to the FLe in framing bias specifically? It seems to me that this hypothesis is the flip-side of what the other two hypothesis are claiming, and a clearer explanation is needed. The cognitive overload hypothesis, as presented here, is based on studies investigating the FLe on moral decision-making. Could the authors justify why they believe that the FLe on framing bias, moral decision-making, and even risk taking or economical all have the same underpinnings? I do not think that this is established yet, and given the differences in the different types of decision making, it may be premature to assume that all FLe arise from the same mechanisms.

91-93: Could the authors expand on this?

116-117: What is meant with “as anonymous reviewers suggest”?

128-130: The authors could be a bit more general when writing about Spanish being the heritage language. There are several other heritage languages, but the way it is presented here it makes it look like only Spanish is. The sentence should be rephrased

134-144: The term “emotional resonance” should be defined and explained in more detail, and information about how emotional resonance was measured should be moved to the Methods.

166-169: This could be rephrased and explained more clearly.

197-200: Unless the authors cannot clearly explain the connection between framing bias and moral decision making regarding the cognitive overload hypothesis, I do not think that they can make this hypothesis. More utilitarian decisions and reduced framing bias in a second language are two different effects using different types of tasks.

Methods

248: provide standard deviation as well

248-249: what is meant with “They were matched on age and socioeconomic status, as well as on the geographic area of origin”? Matched with whom? Or do the authors mean that there was a good spreading for origin and socioeconomic status? If so, provide data.

252: could you provide more information about this questionnaire? Was it questions that the authors put together themselves, or is it an established and validated questionnaire? What were the questions like? What were the answer options? Etc…

261: An objective measure of English would have made the study much stronger, and would have allowed to statistically confirm that the sample was more proficient in their L2 than in their L1. Although the authors asked the participants to rate their language skills, these types of measures are not as reliable. Since it is crucial to the hypothesis that the participants are more proficient in their L2 than in their L1, I do not think that self-reported proficiency is robust enough.

263-265: what was the scale participants were answering on? Did you add the scores from the different questions, or calculate a mean? More information is needed.

266: specify which type of t-test was performed, and provide information about the effect size.

269-271: this is quite relative, isn’t? Since there is no equivalent test in English, it is impossible to determine if the actual proficiency in L2 was higher than in L1. The hypothesis is based on the actual proficiency of the participants, not on their self-perceived proficiency. Yet, only the perceived proficiency is measured in both languages. This is actually an important issue since the actual proficiency is based on assumptions. Higher proficiency in English has not been robustly established. Participants' language skills should have been tested in both languages and statistical analysis should have been performed on the results to determine whether participants truly are more proficient in their L2 (English) than in their L1 (Spanish). Although self-ratings suggest that this is the case, they are not an objective measure of language proficiency.

273-287: this is not completely clear. Could you explain in more detail? How many tasks of each type did the participants perform? And how many of each type/language? How was randomisation/counterbalancing achieved? How were participants assigned to the groups presented in Table 3? Were the answer options also randomised/counterbalanced? More information needs to be provided here.

295-298: what is the norming study? What was done, and how? What scales were used? What was measured and how?

298-301: It appears that the emotional sentences all were personal (directed to the reader), while the non-emotional sentences were not. Was it the case, or were there emotional impersonal/non-emotional personal sentences as well? If not, why was this choice made?

303-307: is this the description of the norming study named on line 295? Or is it the participants of the current studies who rated the sentences. Explain in more detail why (I suspect that it was to control for their perceived emotionality in L1 and L2, but it is not clear from the text). Were the sentences presented randomly?

397: why choose this task? There are other types of tasks that include framing without the emotional component. This would have provided a better control than this type of task.

435: do I understand correctly that 4 emotional and 2 non-emotional tasks were presented. Why was there an uneven number of emotional and non-emotional tasks? This is an issue considering that the use of emotional and non-emotional tasks was an important aspect of the design in order to determine whether emotionality is an underlying mechanism or not. Table 4 should be explained better. It is unclear how many tasks were presented in the loss vs gain framing. Additionally, I understand that the exclusion of non HS participants led to asymmetries, but some scenarios were answered to by very few participants. Since there is not enough information about the design of the task, it is difficult to judge the suitability of the design that was used.

Results

444-457: the way I understand it, knowing whether participants perceived their L1 and L2 as equally emotional is something that should have been controlled for in the same way that proficiency was. Therefore, I do not understand why these results and this information appears in this section. It would have been important to establish beforehand that the participants were more fluent, but equally emotional, in their L2. Furthermore, the way that perceived level of emotion in L1 and L2 is measured raises some questions. It is unclear which variables the Mann-Whitney test was performed on, and the actual result of the analysis is missing. Figure 1 does not establish that emotionality was equal between the two languages since there is a lack of statistical information. The analysis suggest even that valence was rated as higher for some English sentences. Were both languages really equivalent in terms of emotion? More statistical analyses/information is needed. On a related point, why was this specific task chosen? Wouldn’t a task such as an emotional Stroop task in L1 and L2 have been a better way to determine whether participants truly experience both languages equally in terms of emotion? Again, given the importance of perceived emotionality for the hypothesis, it is crucial to establish that there is no difference between L1 and L2 in a robust and reliable way.

460: I am not sure why the authors chose to compare their results to this specific study by Costa et al. This should be explained and justified.

470-483: I do not understand the choice of statistical analysis. Could this be justified and explained too? Since you have count-data, why not perform a chi-2 test (which is what is usually done in this type of studies)? Or did I miss something important which explains why you chose to perform this specific analysis instead?

485: Was proficiency part of the main hypothesis? Why perform this analysis? And why only in Spanish? Was it the perceived proficiency score that was used, or the language test score?

494-524 and 525-543: I have the same questions as for lines 460-485. The reasons why those choices were made are not clear to me, and the analysis do not seem to suit the hypothesis, nor the type of data that you have.

557: The symbol for percentage (%) should not be included in the table itself.

559: more information is needed on the analysis and statistical results

565: still unclear why Spanish proficiency was used as a predictor.

593: in general, the choice of analyses in the results section needs to be explained better, and more statistical results must be provided. It is not sufficient to provide a p-value for instance.

Discussion

Due to the limitations presented above, I do not believe that the conclusions that are drawn are robust. Furthermore, I find it difficult to comment the discussion due to the limitations in the design and analyses.

605-617: this is the main strength of the study and is very promising

618-622: this should have been something that is controlled for just like language proficiency was rather than tested in the results

659-664: I am not sure what is meant here. Could this be clarified?

665-680: I think that these conclusions are premature given the flaws in the design and the analyses. Can you really draw these conclusions?

699-701: Again, since the difference in proficiency was not robustly established, can you be sure about this?

Other comments:

Formalia: check throughout the text for citing formalia. For instance, ampersand (&) should be used when a reference is given within parenthesis, and a comma (,) should be used before the year of the publication (e.g., “(Tversky & Kahneman, 1981)” not “(Tversky and Kahneman 1981). Furthermore, when several references are provided in parenthesis, they should be in alphabetical order based on the first author’s name, and separated by a semicolon (;) (e.g., “(Kahneman, 2003, 2011; Kahneman & Frederick, 2007; Tversky & Kahneman, 1986, 1991)”, not “(Tversky and Kahneman 1986, 1991, Kahneman 2003, 38 2011, Kahneman and Frederick 2007)”.

Have a look at the reference list as well, which is not formatted properly.

Statistics symbols (such as M and SD) should be written in italics.

Author Response

Thank you for your detailed and thoughtful comments. Below are the responses 

  • 48-53: the authors refer to a study by Costa et al., 2014 where they suggest that lower emotionality leads to more utilitarian effects. However, this study investigated moral decision making, not the framing bias. These are two different decision making tasks that share common mechanism but where important differences are found. The authors need to present evidence that shows that framing bias tasks and moral decision making tasks are equivalent.

The study we are referring to here is not one on moral dilemmas by Costa, Foucart, Hayakawa, Aparici, Apesteguia, Heafner, & Keysar published in PLOS 1 in 2014, but the one on the FLe on heuristic biases, specifically on loss aversion, risk aversion, and ambiguity aversion, by Costa, Foucart, Arnon, Aparici, & Apesteguia, published in Cognition in 2014. We made sure the in-text references clearly reflect this distinction throughout the paper.

We agree that the mechanisms in moral decision making and in heuristic biases may involve important differences.

  • 75-76: the authors use the term “utilitarian” and “more rational” as synonyms, but that is not really true. Utilitarian refers to the greater good of the many rather than the few, which is not necessarily more rational. In moral decision making tasks, such as the Footbridge dilemma, it is indeed more rational to sacrifice the life of one person to save the life of five persons, but the terms are not synonyms.

We agree that “utilitarian” and “more rational” are not synonyms, and we changed our wording accordingly throughout the text.  

  • 75-93: Here again the authors present evidence related to moral decision making.

We did not mean to claim that Greene at al. (2008) addressed the FLe; that was a poorly constructed sentence and thus was removed from the text.

  • It appears to me that this theoretical point of view, as presented by Hayakawa et al. 2017 in relation to the FLe, is related to the first two hypothesis. Indeed, they write that what they find suggests that the FLe in moral decision making is likely due to reduced emotionality rather than increased deliberation.

Hayakawa et al. (2017) initially set out to compare two hypotheses that potentially account for the FLe – the blunted-deontology account and the heightened-utilitarianism account. These are similar (although we agree that they are not the same) to the reduced emotionality hypothesis and cognitive enhancement hypothesis discussed in our study. However, in three of their six experiments, Hayakawa el al. found that foreign language did not increase utilitarian responding but in fact decreased it. That is, answering the moral dilemmas in FL led to less utilitarian results. Based on this finding, the authors suggested that, “One possibility is that decreased utilitarianism among foreign-language users resulted from an increase in cognitive load. Using a foreign language can be cognitively demanding, especially for speakers who are not highly proficient (Plass, Chun, Mayer, & Leutner, 2003), and cognitively demanding tasks impair utilitarian responding (Greene et al., 2008).” That is, the data in Hayakawa et al. study did not support the heightened-utilitarianism account, and the authors proposed an alternative explanation by suggesting that the higher cognitive load actually impairs one’s ability to care about maximizing benefit.

  • Indeed, the authors of this manuscript write “That is, if there is a FLe, it is actually the opposite of what has been assumed in the earlier studies - an increased cognitive load does not enhance but rather affects one’s ability to provide a utilitarian answer”. However, this is not what the FLe in moral decision making finds, is it? Usually, it finds that utilitarian responses increasewhen using a second language, not diminish.

We thank the reviewer for the comment. Early research did show that utilitarian responses increase when using a FL. However, recent studies using more suitable methodologies (e.g., the CNI model or Process Dissociation Approach) have shown that the FLe, in addition to blunting deontological inclinations, actually leads to decreased utilitarian responding – people don’t seem to care as much about maximizing benefit when they read dilemmas in their FL (Hayakawa et al., 2017, Muda et al, 2018, Bialek et al, 2019).

  • Furthermore, this is related to moral decision making, not framing bias. Could the authors clarify this point in relation to the FLe in framing bias specifically? It seems to me that this hypothesis is the flip-side of what the other two hypothesis are claiming, and a clearer explanation is needed. The cognitive overload hypothesis, as presented here, is based on studies investigating the FLe on moral decision-making. Could the authors justify why they believe that the FLe on framing bias, moral decision-making, and even risk taking or economical all have the same underpinnings? I do not think that this is established yet, and given the differences in the different types of decision making, it may be premature to assume that all FLe arise from the same mechanisms.

We address this comment in lines 58-67 of the revised manuscript by discussing a theoretical foundation as to why the different types of decision-making involve similar mechanisms.

  • 91-93: Could the authors expand on this?

Addressed in lines 896-901

  • 116-117: What is meant with “as anonymous reviewers suggest”?

This is an error and was removed.

  • 128-130: The authors could be a bit more general when writing about Spanish being the heritage language. There are several other heritage languages, but the way it is presented here it makes it look like only Spanish is. The sentence should be rephrased

Addressed in lines 142-146 of the revised manuscript.

  • 134-144: The term “emotional resonance” should be defined and explained in more detail, and information about how emotional resonance was measured should be moved to the Methods.

Addressed in lines 155-156 of the revised manuscript.

  • 166-169: This could be rephrased and explained more clearly.

Addressed in lines 189-197 of the revised manuscript.

  • 197-200: Unless the authors cannot clearly explain the connection between framing bias and moral decision making regarding the cognitive overload hypothesis, I do not think that they can make this hypothesis. More utilitarian decisions and reduced framing bias in a second language are two different effects using different types of tasks.

While we agree that moral decision making and the role of heuristics in decision making are different and may involve different underlying mechanisms, we believe that increasing cognitive load by presenting tasks in the FL may affect the two in similar ways. This is because both involve conscious cognitive effort, which increases if tasks are presented in a FL or less proficient language. This is because processing in one’s L2/FL is cognitively more costly, e.g., depletes working memory capacity sooner, etc.

For the same reasons, we believe that for cognitive load hypothesis can be made in our paper. We note that Costa et al (2014) explicitly discuss the cognitive load hypothesis with respect to decision making and heuristic biases: “A second relevant factor is the cognitive load. Under conditions of high cognitive load participants’ decisions tend to be more affected by heuristic biases (Benjamin, Brown, & Shapiro, 2006; Forgas, Baumeister, & Tice, 2009; Whitney, Rinehart, & Hinson, 2008). That is, when cognitive load taxes System 2, the rational processor cannot check or control the intuitive answers given by System 1. Hence, to the extent that reading in a FL increases cognitive load, one might expect heuristic biases to affect participants’ responses to a larger extent when the problem is set in a FL.”

Methods

  • 248: provide standard deviation as well

Added in line 296.

  • 248-249: what is meant with “They were matched on age and socioeconomic status, as well as on the geographic area of origin”? Matched with whom? Or do the authors mean that there was a good spreading for origin and socioeconomic status? If so, provide data.

This sentence was included in error and has been removed.

  • 252: could you provide more information about this questionnaire? Was it questions that the authors put together themselves, or is it an established and validated questionnaire? What were the questions like? What were the answer options? Etc…

We have added Appendix 2 to address this comment.

  • 261: An objective measure of English would have made the study much stronger, and would have allowed to statistically confirm that the sample was more proficient in their L2 than in their L1. Although the authors asked the participants to rate their language skills, these types of measures are not as reliable. Since it is crucial to the hypothesis that the participants are more proficient in their L2 than in their L1, I do not think that self-reported proficiency is robust enough.

While we agree that self-reported proficiency is not a robust measure, we also provide objective information regarding our participants’ linguistic background that strongly suggests that English was their more proficient language. Their schooling was done mostly in English through secondary school and beyond (74%), time spent in English vs. Spanish-speaking countries was 81% vs. 23% respectively, and their own preference in addressing someone who speaks both English and Spanish was 72% vs. 28%, all suggesting that the dominant social language is English.

  • 263-265: what was the scale participants were answering on? Did you add the scores from the different questions, or calculate a mean? More information is needed.

This additional information has been added in line 312.

  • 266: specify which type of t-test was performed, and provide information about the effect size.

This is now included in line 315.

  • 269-271: this is quite relative, isn’t? Since there is no equivalent test in English, it is impossible to determine if the actual proficiency in L2 was higher than in L1. The hypothesis is based on the actual proficiency of the participants, not on their self-perceived proficiency. Yet, only the perceived proficiency is measured in both languages. This is actually an important issue since the actual proficiency is based on assumptions. Higher proficiency in English has not been robustly established. Participants' language skills should have been tested in both languages and statistical analysis should have been performed on the results to determine whether participants truly are more proficient in their L2 (English) than in their L1 (Spanish). Although self-ratings suggest that this is the case, they are not an objective measure of language proficiency.

Please, see our answer to comment to lines 261 above.

  • 273-287: this is not completely clear. Could you explain in more detail? How many tasks of each type did the participants perform? And how many of each type/language? How was randomisation/counterbalancing achieved? How were participants assigned to the groups presented in Table 3? Were the answer options also randomised/counterbalanced? More information needs to be provided here.

We have moved this paragraph later in the text after the description of the tasks, which should clarify the design.

  • 295-298: what is the norming study? What was done, and how? What scales were used? What was measured and how?

We added more details about the norming study in lines 328-342.  

  • 298-301: It appears that the emotional sentences all were personal (directed to the reader), while the non-emotional sentences were not. Was it the case, or were there emotional impersonal/non-emotional personal sentences as well? If not, why was this choice made?

We adopted the Emotional Phrases Task as used in Caldwell-Harris & Ayçiçegi-Dinn (2009) and Harris (2003, 2004), where all emotional sentences are directed to the reader. 

  • 303-307: is this the description of the norming study named on line 295? Or is it the participants of the current studies who rated the sentences. Explain in more detail why (I suspect that it was to control for their perceived emotionality in L1 and L2, but it is not clear from the text). Were the sentences presented randomly?

The text in lines 328-342 now clarifies the issue.

  • 397: why choose this task? There are other types of tasks that include framing without the emotional component. This would have provided a better control than this type of task.

Cognitive Reflection Test has been used extensively to test the effect of cognitive load on performance, and cognitive load is one of the main potential predictors of FLe. We did not aim to test only framing bias, but rather sought to test different cognitive biases that emerge during decision-making. For example, the ticket/money problem considers the biases that humans exhibit during mental accounting of economic outcomes, the Holt-Laury lotto tests decision-making under risk and uncertainty, etc. (the full description of the type of bias is in the methods section). The only framing bias task we have used is Asian Disease task.  

  • 435: do I understand correctly that 4 emotional and 2 non-emotional tasks were presented. Why was there an uneven number of emotional and non-emotional tasks? This is an issue considering that the use of emotional and non-emotional tasks was an important aspect of the design in order to determine whether emotionality is an underlying mechanism or not.

Emotionality was not an independent variable in any of the statistical tests, so we did not compare emotional vs. non-emotional tasks directly. The choices across emotional and non-emotional tasks are not directly comparable.

  • Table 4 should be explained better. It is unclear how many tasks were presented in the loss vs gain framing. Additionally, I understand that the exclusion of non HS participants led to asymmetries, but some scenarios were answered to by very few participants. Since there is not enough information about the design of the task, it is difficult to judge the suitability of the design that was used.

Table 4 has been changed to address the comment.

  • 444-457: the way I understand it, knowing whether participants perceived their L1 and L2 as equally emotional is something that should have been controlled for in the same way that proficiency was. Therefore, I do not understand why these results and this information appears in this section. It would have been important to establish beforehand that the participants were more fluent, but equally emotional, in their L2.

Emotionality rating results have been moved to section 2.1. Section 2.1 also includes further results from the background questionnaire (now included as an appendix 1) supporting the notion that participants were dominant in English.

  • Furthermore, the way that perceived level of emotion in L1 and L2 is measured raises some questions. It is unclear which variables the Mann-Whitney test was performed on, and the actual result of the analysis is missing.

We have added clarification and actual results of the Mann-Whitney test in lines 349-356.

  • Figure 1 does not establish that emotionality was equal between the two languages since there is a lack of statistical information. The analysis suggest even that valence was rated as higher for some English sentences. Were both languages really equivalent in terms of emotion? More statistical analyses/information is needed.

As the text points out, 20 of the 22 sentences had ratings that were not statistically significantly different, and only 2 had statistically significantly different ratings. We also added more information about the norming study that clarifies our procedure for ensuring emotional comparability of the sentences (lines 328-342)

  • On a related point, why was this specific task chosen? Wouldn’t a task such as an emotional Stroop task in L1 and L2 have been a better way to determine whether participants truly experience both languages equally in terms of emotion? Again, given the importance of perceived emotionality for the hypothesis, it is crucial to establish that there is no difference between L1 and L2 in a robust and reliable way.

This task was adopted from previous studies in the literature (Caldwell-Harris et al., 2009; Harris 2003, 2004). In addition, as we point out in the text, numerous studies on language and emotionality have shown that early bilinguals, like the ones in our study, have equal emotional resonances in their languages.

  • 460: I am not sure why the authors chose to compare their results to this specific study by Costa et al. This should be explained and justified.

Results have been reframed independently of Costa et al.’s study throughout the section.

  • 470-483: I do not understand the choice of statistical analysis. Could this be justified and explained too? Since you have count-data, why not perform a chi-2 test (which is what is usually done in this type of studies)? Or did I miss something important which explains why you chose to perform this specific analysis instead?

The use of a linear regression allowed us to explore causality between language and Framing, which a regular chi-2 test usually does not.

  • 485: Was proficiency part of the main hypothesis? Why perform this analysis? And why only in Spanish? Was it the perceived proficiency score that was used, or the language test score?

The cognitive load and cognitive enhancement hypotheses are both based on the observed finding that processing in one’s FL/L2 is cognitively more costly, and where lower proficiency is correlated with higher cognitive load. Both cognitive load and cognitive enhancement hypotheses, therefore, would predict that in FL learners FLe is modulated by language of presentation (L1 vs. FL) and by proficiency in the FL. In our HS speakers, the L1 is the less proficient language, due to attrition and other processes that take place in the heritage languages (on this point, see Montrul & Polinsky, 2019). Thus, only the L1 was used to test the effect of proficiency (the test score, now clarified in the text), because we assumed that English was the dominant language and therefore proficiency would be at ceiling. These participants have had most of their schooling in English, they have been exposed to English since early childhood, and there is no a priori reason to think their English is not native-like. Please see lines 299-317 for more info supporting this. As far as we can tell, this is the standard assumption in all of the field of heritage language studies, although perhaps that is an ungranted assumption that should be revised.

  • 494-524 and 525-543: I have the same questions as for lines 460-485. The reasons why those choices were made are not clear to me, and the analysis do not seem to suit the hypothesis, nor the type of data that you have.

As far as we can tell, generalized linear regressions can be applied to count data.

  • 557: The symbol for percentage (%) should not be included in the table itself.

% has been taken out of the table itself

  • 559: more information is needed on the analysis and statistical results

We have added significance for the predictor variable

  • 565: still unclear why Spanish proficiency was used as a predictor.

Please, see our answer to comment on lines 485 above.

  • 593: in general, the choice of analyses in the results section needs to be explained better, and more statistical results must be provided. It is not sufficient to provide a p-value for instance.

We thank the reviewer for this comment, but it is not clear what kind of explanation regarding the choice of analyses is requested, given that linear regression models are fairly standard procedures. It would be helpful to know what additional statistical information would be useful to interpret results, particularly when the effects were not significant.

  • 618-622: this should have been something that is controlled for just like language proficiency was rather than tested in the results

We used the Emotional Phrases Task to control for emotionality. While this, as we acknowledge in our discussion, is not the most reliable measure, many studies on FLe on decision-making do not involve any measures of emotionality at all. In fact, if we are not mistaken, the only study that does is Miozzo et al. (2020), who also used the same Emotional Phrases Task in their study on Italian-Venetian and Italian-Bergamasque bilinguals. 

  • 659-664: I am not sure what is meant here. Could this be clarified?

Different cognitive biases have been identified and studied in monolingual literature. These have not been studied specifically in heritage speakers. While there is no reason to suspect that human cognitive biases do not apply to heritage speakers, this is an empirical question that needs to be addressed. More importantly for the purposes of the present study, however, before we could establish whether there is a language effect on those biases in heritage speakers, we needed to establish whether there are biases in this bilingual population at all. As our results show, we did find biases in HS speakers across our decision-making tasks.  We have added this clarification to the discussion in lines 784-787.

  • 665-680: I think that these conclusions are premature given the flaws in the design and the analyses. Can you really draw these conclusions?

We believe that we can. We also do acknowledge alternative interpretations in several parts of the discussion, e.g., in lines 838-861.

  • 699-701: Again, since the difference in proficiency was not robustly established, can you be sure about this?

As we explain above, we did measure Spanish proficiency using both a self-report and an objective measure. As for English proficiency, we provided information that indicates that English is the dominant language in our HS speakers.

  • Other comments: Formalia: check throughout the text for citing formalia. For instance, ampersand (&) should be used when a reference is given within parenthesis, and a comma (,) should be used before the year of the publication (e.g., “(Tversky & Kahneman, 1981)” not “(Tversky and Kahneman 1981). Furthermore, when several references are provided in parenthesis, they should be in alphabetical order based on the first author’s name, and separated by a semicolon (;) (e.g., “(Kahneman, 2003, 2011; Kahneman & Frederick, 2007; Tversky & Kahneman, 1986, 1991)”, not “(Tversky and Kahneman 1986, 1991, Kahneman 2003, 38 2011, Kahneman and Frederick 2007)”.

We have double-checked the journal’s house style (Chicago) and have adopted it throughout.

  • Have a look at the reference list as well, which is not formatted properly.

Addressed throughout the reference page.

  • Statistics symbols (such as and SD) should be written in italics.

Addressed where relevant.

Round 2

Reviewer 3 Report

Thank you to the authors for their revisions and improvements. I still have major concerns with the manuscript that should be addressed before reconsideration:

  1. Measurement of proficiency: I understand the authors’ argument for claiming that their participants were more proficient in their L2 than in their L1, which I agree is likely. On the other hand, bilingualism is a broad and ill-defined concept that is operationalised in a panoply of different ways in research. Therefore, one cannot assume that the bilinguals that they study are the same type of bilinguals that others have studied. Although self-ratings have been and continue being used in bilingualism research, they are also highly criticised. Thus, this is a limitation in this study. I understand that this part of the study cannot be changed after data collection, but the authors should nonetheless be careful about the conclusions that are drawn, and this methodological limitation should be acknowledged and addressed in the Discussion. In my opinion, this would increase the scientific value and transparency of the study. Perhaps it would be useful to present it as a way to improve future studies on the subject?
  2. Although the Emotional Phrases Task has been used in previous studies, it is not used with the same procedure in this study. This is not necessarily essential, but it is not sufficient to refer to previous studies, especially since the procedure is different. Most importantly perhaps, the way that the data is analysed still seems strange to me. I cannot understand why individual analysis were made for each sentence instead of a 2x2 ANOVA. This would be more appropriate (and more in line with previous studies) than several individual analyses. It would provide more convincing evidence that the level of perceived emotion is equal across languages (and ANOVA analyses are generally considered to be robust enough to be used with a 5 steps scale). Also, it would be interesting to have information on why a between-subject was chosen for this task (this differs also from how the task has been administered in previous studies).
  3. Regression analyses: If the data is normally distributed, it is possible to use on count data. However, since the dependent variable can only be one of two options, using a Poisson or Negative binomial is more appropriate. I would recommend the following publications and doing the necessary adjustments:

Gardner, W., Mulvey, E.P., and Shaw, E.C (1995). “Regression Analyses of Counts and Rates: Poisson, Overdispersed Poisson, and Negative Binomial Models”, Psychological Bulletin, 118, 392-404.

Long, J.S. (1997). Regression Models for Categorical and Limited Dependent Variables, Chapter 8. Thousand Oaks, CA: Sage Publications.

Or perhaps even a Wald test would be better depending on the question:

Fox, J. (1997) Applied regression analysis, linear models, and related methods. Thousand Oaks, CA: Sage Publications.

Johnston, J. and DiNardo, J. (1997) Econometric Methods Fourth Edition. New York, NY: The McGraw-Hill Companies, Inc.

Note also that causality is determined by the type of design, not by the type of analysis. With this said, the purpose of the study appears to be to detect whether a FLe occurs in a HS population in different types of decision-making scenarios to test three possible explanations to the FLe. Based on the argumentation that the authors present in the introduction and on the design, simply analysing the data of the different tasks with chi-2 tests to detect whether a FLe occurs would fit the design better and give a clear answer to the hypothesis presented in table 2. I cannot see why proficiency should be used as a predictor, or how it is related to the hypothesis presented in table 2. On the other hand, if the authors would like to investigate whether the level of proficiency predicts the choice that is made, then a regression is more appropriate. In this case, the introduction should be changed to fit the purpose and analyses. It all depends on what the authors are aiming to do here.

On a side note, do the authors mean that the self-bias (rather than FLe) is caused by foreignness in the following sentence: “The results showed a reduction of self-bias in the non-native foreign (Eng-1175 lish) language, but not in the non-native local language (Basque), which indicates that the FLe is caused by the foreignness, not non-nativeness of a language.”? Also, Table 3 simply repeats information that is in the text. If the information is easier to understand with a table, I would suggest referring to the table and removing the data from the text. And if the information in the text is clear as it is, the table is redundant.

Author Response

Thank you for taking the time to read this revised version and provide further comments.

  1. Measurement of proficiency: I understand the authors’ argument for claiming that their participants were more proficient in their L2 than in their L1, which I agree is likely. On the other hand, bilingualism is a broad and ill-defined concept that is operationalised in a panoply of different ways in research. Therefore, one cannot assume that the bilinguals that they study are the same type of bilinguals that others have studied. Although self-ratings have been and continue being used in bilingualism research, they are also highly criticised. Thus, this is a limitation in this study. I understand that this part of the study cannot be changed after data collection, but the authors should nonetheless be careful about the conclusions that are drawn, and this methodological limitation should be acknowledged and addressed in the Discussion. In my opinion, this would increase the scientific value and transparency of the study. Perhaps it would be useful to present it as a way to improve future studies on the subject?

We have added a paragraph (lns. 1448-1455) with some discussion on this topic

2. Although the Emotional Phrases Task has been used in previous studies, it is not used with the same procedure in this study. This is not necessarily essential, but it is not sufficient to refer to previous studies, especially since the procedure is different. Most importantly perhaps, the way that the data is analysed still seems strange to me. I cannot understand why individual analysis were made for each sentence instead of a 2x2 ANOVA. This would be more appropriate (and more in line with previous studies) than several individual analyses. It would provide more convincing evidence that the level of perceived emotion is equal across languages (and ANOVA analyses are generally considered to be robust enough to be used with a 5 steps scale). Also, it would be interesting to have information on why a between-subject was chosen for this task (this differs also from how the task has been administered in previous studies).

We now have results from an ANOVA. Notice that this required us to treat emotionality as a categorical value and divide the sentences in two groups (ln 490-494).

3. Regression analyses: If the data is normally distributed, it is possible to use on count data. However, since the dependent variable can only be one of two options, using a Poisson or Negative binomial is more appropriate.

It seems like the suggestion is based on the idea that the number of yes/no answers should be treated count data, but after reading the suggested references and further consulting with statistical experts, we decided that while it would be interesting to treat binary responses as counts, it is also a standard statistical analysis to submit them to a logit regression. 

 I cannot see why proficiency should be used as a predictor, or how it is related to the hypothesis presented in table 2. On the other hand, if the authors would like to investigate whether the level of proficiency predicts the choice that is made, then a regression is more appropriate. In this case, the introduction should be changed to fit the purpose and analyses. It all depends on what the authors are aiming to do here.

Two of the hypotheses in table 2 rely on the effect of slower processing in a second language on reasoning. The Cognitive enhancement hypothesis assumes that effect is positive because it allows for more rational processes to apply, and the cognitive overload assumes the opposite: processing load impairs rational processes. In both cases, what is at stake is how efficient processing is, and one of the factors related to efficient processing is proficiency, that is why we are interested in seeing whether changes in proficiency have an effect on framing biases.